# Strategies to package recombinant Adeno-Associated Virus expressing the N-terminal gasdermin domain for tumor treatment

Yuan Lu [1,2,6], Wenbo He[1,2,6], Xin Huang[1,2], Yu He [1,2], Xiaojuan Gou[1,2], Xiaoke Liu[1,2], Zhe Hu[1], Weize Xu[1,2], Khaista Rahman[1,2], Shan Li [3], Sheng Hu[4✉], Jie Luo[5✉] & Gang Cao [1,2,3✉]

Pyroptosis induced by the N-terminal gasdermin domain (GSDM$^{NT}$) holds great potential for anti-tumor therapy. However, due to the extreme cytoxicity of GSDM$^{NT}$, it is challenging to efficiently produce and deliver GSDM$^{NT}$ into tumor cells. Here, we report the development of two strategies to package recombinant adeno-associated virus (rAAV) expressing GSDM$^{NT}$: 1) drive the expression of *GSDM$^{NT}$* by a mammal specific promoter and package the virus in Sf9 insect cells to avoid its expression; 2) co-infect rAAV-Cre to revert and express the double-floxed inverted *GSDM$^{NT}$*. We demonstrate that these rAAVs can induce pyroptosis and prolong survival in preclinical cancer models. The oncolytic-viruses induce pyroptosis and evoke a robust immune-response. In a glioblastoma model, rAAVs temporarily open the blood-brain barrier and recruit tumor infiltrating lymphocytes into the brain. The oncolytic effect is further improved in combination with anti-PD-L1. Together, our strategies efficiently produce and deliver GSDM$^{NT}$ into tumor cells and successfully induce pyroptosis, which can be exploited for anti-tumor therapy.

[1] State Key Laboratory of Agricultural Microbiology, Hubei Hongshan Laboratory, Huazhong Agricultural University, 430070 Wuhan, China. [2] College of Veterinary Medicine, Huazhong Agricultural University, 430070 Wuhan, China. [3] College of Biomedicine and Health, Huazhong Agricultural University, 430070 Wuhan, China. [4] Department of Medical Oncology, Hubei Cancer Hospital, Huazhong Agricultural University, Wuhan, China. [5] Taihe Hospital, Hubei University of Medicine, Shiyan 442000 Hubei, China. [6] These authors contributed equally: Yuan Lu, Wenbo He. ✉email: ehusmn@163.com; luojie_001@126.com; gcao@mail.hzau.edu.cn

The immunosuppressive feature of the tumor micro-environment (TME) allows tumor cells to escape the immune clearance[1–3]. Recent studies have demonstrated that pyroptosis can directly kill tumor cells, and more importantly, trigger a strong antitumor immune response, which recruits a large number of lymphocytes and destroys the TME[4–6]. Pyroptosis is a programmed cell death caused by the pore-forming N-terminal domain of gasdermin (GSDM^NT). It is accompanied by the release of inflammatory substances, which leads to a strong inflammatory response[7–10]. The combined treatment with pyroptosis and the immune checkpoint PD1 monoclonal antibody has been proposed to further improve the oncolytic efficacy[5,6]. While pyroptosis holds a great promise for cancer therapy, it is still challenging to efficiently produce and deliver GSDM^NT into tumor cells.

Several attempts have been taken to express full-length GSDME/B in the tumor cells[4,5] or deliver GSDMA3 (N + C) into tumor cells using the bio-orthogonal system[6]. However, these approaches under current conditions are still inefficient and costly. Considering the low immunogenicity, high gene delivery efficacy, and replication deficiency characteristics[11], it may be ideal to use recombinant adeno-associated viruses (rAAVs) to deliver and express GSDM^NT gene in the tumor cells. rAAV has been widely used in clinical gene therapy for many diseases, such as hemophilia B[12] and Leber congenital amaurosis[13,14]. Moreover, rAAV has strong penetrability to solid tumors and has already been applied in tumor therapies[15,16]. However, GSDM^NT is highly toxic to the cells[17]; its intracellular expression can lead to pyroptosis during AAV packaging and thus poses a challenge to delivery GSDM^NT by AAV. While it is a good strategy to drive the expression of GSDM^NT in rAAV by cell-specific promoter, as it has been successfully demonstrated with a Schwann-cell specific promoter for Schwannoma cancer therapy[18], some cell-specific promoters cannot completely avoid the leakage expression of GSDM^NT during AAV packaging process, which may affect the titer of the virus. Novel strategies are therefore desperately needed to package and deliver GSDM^NT.

In the current communication, we report the development of an rAAV system to express and deliver GSDM^NT into cancer cells. The rAAVs can induce pyroptosis that subsequently stimulate immune responses and temporarily open the blood–brain barrier (BBB). These strategies are likely to have an impact on central and peripheral antitumor therapies in the future.

## Results

### Development of recombinant AAV packaging strategies to express GSDMD^NT.
As baculovirus/Sf9 cell system has been developed to package rAAV, we hypothesized that this insect packaging system with a mammal-specific promoter may avoid the severe GSDM^NT cytotoxicity during the process of virus packaging (Fig. 1a). Thus, we screened several promoters and selected a mammal CBA (mCBA) promoter (Supplementary Fig. 1) to drive GSDMD^NT expression, of which the transcription activity in Sf9 insect cells was firstly verified using enhanced green fluorescent protein (eGFP). As shown in Fig. 1b, a negligible expression level of eGFP was observed in the Sf9 insect cells as compared to the CMV and CAG promoters. To test the leaky expression of GSDMD^NT driven by mCBA in Sf9 cells, two recombinant Baculovirus plasmids (Bacmids) carrying the CMV-GSDMD^NT and mCBA-GSDMD^NT were constructed and transfected into Sf9 cells to package GSDMD^NT-expressing baculoviruses, respectively. It is found that the titer of rBac-CMV-GSDMD^NT is significantly lower than that of the rBac-mCBA-GSDMD^NT ($5.56 \times 10^6$ vs $1.74 \times 10^8$ pore-forming unit (PFU)/ml). The lower titer of rBac-CMV-GSDMD^NT is most likely due to the leaky expression of GSDMD^NT

under CMV promoter in Sf9 cells which caused the cytotoxicity to Sf9 cells. In contrast, there is hardly any pyroptosis in the mCBA-GSDMD^NT-infected Sf9 cells (Fig. 1c). The activity of mCBA promoter in mammalian cells has also been validated in mammalian cells (Supplementary Fig. 2).

Next, the two-baculovirus packaging system was employed to package rAAV-mCBA-GSDMD^NT by coinfecting Sf9 cells with rBac-mCBA-GSDMD^NT and rBac-AAV/helper baculovirus, which provides the Rep and Cap proteins for rAAV packaging[19] (Supplementary Fig. 3). rAAV-CMV-GSDMD^NT was also packaged by this system for comparison. rAAV-mCBA-GSDMD^NT (named as rAAV-P1) was successfully obtained with high yield ($3.2 \times 10^{10}$ viral genomes (vg)/$10^6$ Sf9 cells), of which the titer is much higher than that of the rAAV-CMV-GSDMD^NT ($5.4 \times 10^7$ vg/$10^6$ Sf9 cells) (Fig. 1d). These data suggest that this strategy can mostly avoid the cytotoxicity of GSDMD^NT during rAAV packaging and successfully produce rAAV-P1 by taking advantage of the minimal transcription activity of mCBA promoter in Sf9 cells. Next, we infected HEK 293T cells with rAAV-P1 to test the expression of GSDMD^NT driven by mCBA promoter in human cells. As shown in Fig. 1g and Supplementary Fig. 5a, infection of rAAV-P1 can indeed induce an obvious pyroptosis in HEK 293T cells.

To completely eliminate the toxicity of GSDMD^NT during rAAV packaging, we developed another strategy using Cre/lox recombinase system to package rAAV-DIO-GSDM^NT (Fig. 1e). In this system, GSDMD^NT was invertedly cloned into rAAV vector flanking with syntropic double Lox sites, which can absolutely avoid the expression of GSDMD^NT (Fig. 1f). Upon coexpression with Cre recombinase, the Cre/lox system can efficiently revert GSDMD^NT and subsequently initiate the expression of GSDMD^NT (Fig. 1e). To test this strategy, we packaged rAAV-Cre and rAAV-DIO-GSDMD^NT in HEK 293T cells (Supplementary Fig. 4). Since rAAV-Cre and rAAV-DIO-GSDMD^NT have to be simultaneously applied for tumor treatment, this system is termed as rAAV-P2.

The rAAV-P2 system was firstly tested in HEK 293T cells, which showed that the coinfection of rAAV-P2 (rAAV-Cre and rAAV-DIO-GSDMD^NT) can indeed induce robust pyroptosis in HEK 293T cells, whereas this was not observed in the HEK 293T cells infected with rAAV-DIO-GSDMD^NT alone (Fig. 1g and Supplementary Fig. 5a). In addition, pyroptosis in the rAAV-P2 treatment group occurred earlier than that of the rAAV-P1 group (Supplementary Fig. 5b, c). Moreover, the lactate dehydrogenase (LDH) release-based cell death assay and the ATP cell viability assay demonstrated that in comparison to the rAAV-P1 treatment group, the rAAV-P2 treatment group caused more cell death (Supplementary Fig. 5d), suggesting that rAAV-P2 can induce stronger pyroptosis than rAAV-P1. This may be due to the lower transduction efficiency of rAAV packaged by Sf9 cells (Supplementary Fig. 6a, b) and the relatively weaker transcription activity of the mCBA promoter (Supplementary Fig. 6c, d).

To further test the pyroptosis effect of rAAV-P1 and rAAV-P2 in tumor cells, four cancer cell lines including HeLa, Hep3B, C6-luc, and 4T1-luc were infected with rAAV-P1 and rAAV-P2, respectively. Both rAAV-P1 and rAAV-P2 induced pyroptosis in all the tumor cells (Fig. 2a–d), indicating the successful expression of GSDMD^NT. Then, LDH release-based cell death assay and ATP cell viability assay were performed to quantitate the lethal effect of rAAV-P1 and rAAV-P2 to the cancer cells. In the control group, the cell death rates of HeLa, C6-luc, and 4T1-luc were 2.2%, 8.5%, and 5.0% respectively, while the counts were 23.2%, 20.48%, and 19.01% in the rAAV-P1-treated group, and 46.92%, 35.5%, and 28.6% in the rAAV-P2-treated group, respectively (Fig. 2e). The ATP cell viability assays also demonstrated that rAAV-P2 treatment caused more cell death compared with rAAV-P1

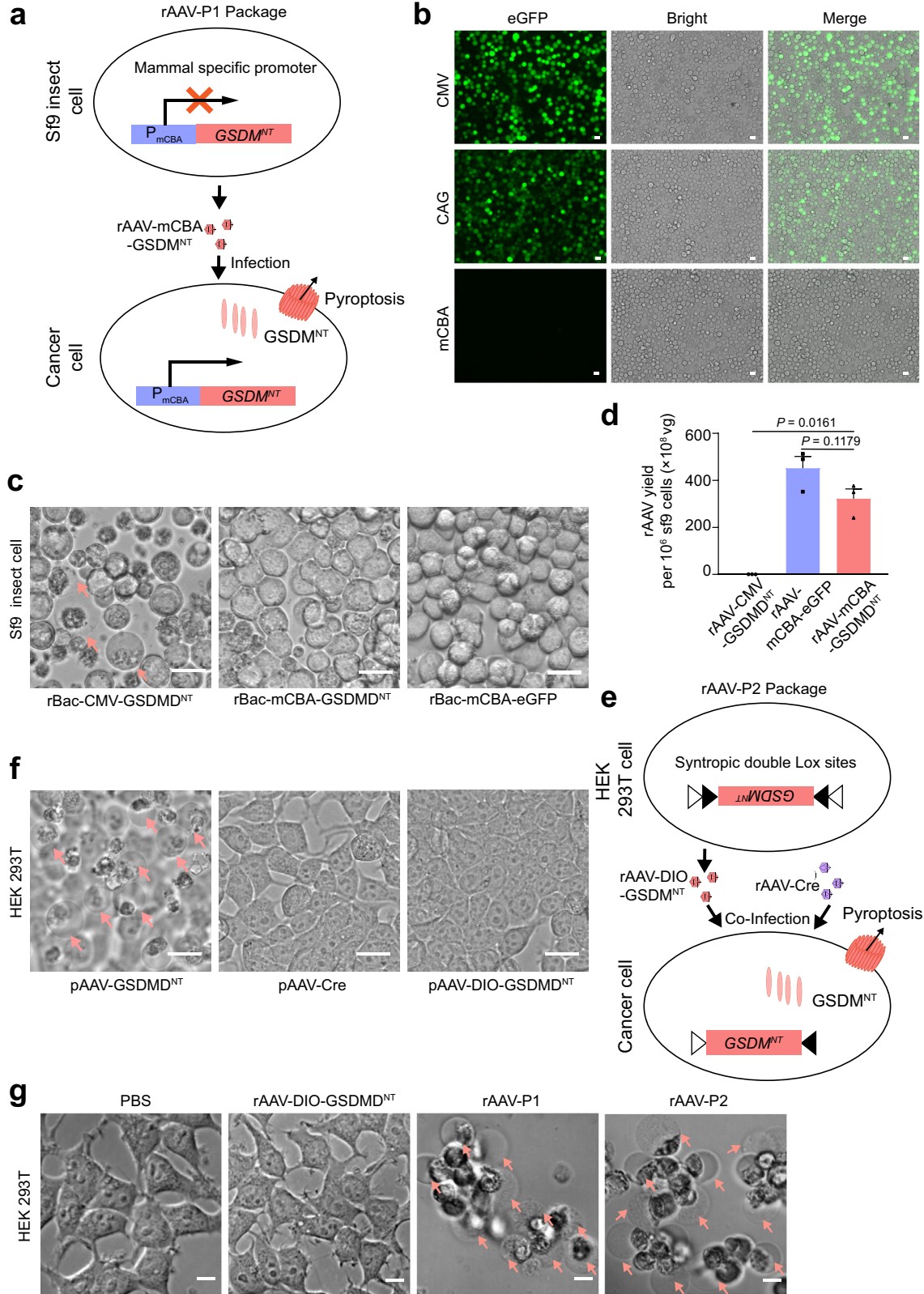

treatment (Fig. 2f). Moreover, we packaged rAAV-expressing GSDMB[NT] (rAAV-P2B) and GSDME[NT] (rAAV-P2E) by Cre/lox system. As shown in Supplementary Fig. 6e, f, rAAV-P2 induced stronger pyroptosis compared with rAAV-P2B and rAAV-P2E. Importantly, it was found that the intratumorally injected rAAV was mostly restricted in the solid tumor tissue (Supplementary

Fig. 7). Thus, rAAV-P2 was chosen as the oncolytic virus for subsequent in vivo tumor treatment experiments.

**rAAV-P2 treatment prolongs the survival time in a rat glioblastoma model.** To evaluate the oncolytic efficacy of rAAV-P2

**Fig. 1 The strategies for packaging recombinant adeno-associated virus (rAAV) expressing GSDMD[NT]. a** Schematic of the strategy for the packaging of rAAV-GSDMD[NT] (rAAV-P1) using the Sf9/rBac system and mammal-specific promoter mCBA. mCBA promoter can hardly drive gene expression in sf9 insect cells. **b** Fluorescence microscopy of Sf9 insect cells showing the expression of eGFP driven by different promoters at 48 h post transfection. Scale bars, 10 μm. **c** Images of the Sf9 insect cells infected with different baculoviruses. Arrows indicate pyroptotic cells. Scale bars, 20 μm. **d** The viral titer of different rAAVs packaged by the Sf9/rBac system. Data were expressed as mean ± s.e.m., from three independent replicates. Two-tailed unpaired *t* test with Welch's correction was used for comparing the difference between two groups as indicated. **e** Schematic of the strategy using Cre/lox system to package rAAV-DIO-GSDMD[NT] (rAAV-ef1α-DIO-GSDMD[NT]). Coinfection of rAAV-Cre can revert the DIO-GSDMD[NT] (rAAV-P2) to induce pyroptosis. **f** Images of HEK 293T cells transfected with pAAV-GSDMD[NT] (pAAV-CMV-DIO-GSDMD[NT]), pAAV-Cre, and pAAV-DIO-GSDMD[NT] (pAAV-ef1α-DIO-GSDMD[NT]), respectively. Arrows indicate pyroptotic cells. Scale bars, 20 μm. **g** Differential interference contrast (DIC) images of HEK 293T cells infected with rAAV-DIO-GSDMD[NT] (rAAV-ef1α-DIO-GSDMD[NT]), rAAV-P1, rAAV-P2, and PBS, respectively. Arrows indicate pyroptotic cells. Scale bars, 20 μm. All data are representative of three independent experiments. Source data are provided as a Source Data file.

treatment for glioblastoma (GBM) in vivo, we established an orthotopic GBM rat model by stereotactic injecting C6 cells expressing luciferase into the left striatum of Wistar rats. Seven days post-tumor implantation, the rats were intratumorally injected with rAAV-P2, rAAV-DIO-GSDMD[NT], or phosphate-buffered saline (PBS) and then subjected to the live animal imaging on day 20 to evaluate GBM progression (Fig. 3a). Compared with the PBS and the rAAV-DIO-GSDMD[NT] control groups, the rats in the rAAV-P2 treatment group showed much mild luciferase signals (Fig. 3b, c), indicating a significant regression of tumor growth. Moreover, rAAV-P2 treatment can significantly attenuate the weight loss caused by GBM (Supplementary Fig. 8a, b) and prolong the average survival time in the GBM rat model (Fig. 3d). Of note, half of the rats in the rAAV-P2 treatment group survived for more than 50 days without any detectable symptoms of GMB (Fig. 3d, e). The histopathology examination of the brain sections on days 35 and 50 post-tumor induction also showed that the rAAV-P2 can effectively inhibit tumor growth (Fig. 3e).

A key reason for the rapid progress of GBM in patients is that the BBB prevents the entrance of the lymphocytes to eliminate tumor cells[20]. To explore the effect of rAAV-P2 treatment on the integrity of BBB, Evans blue dye was intravenously injected into the GBM rat model. As shown in Fig. 3f, g, the brains in the rAAV-P2 treatment group displayed deep Evans blue dye staining, whereas the control group stained only lightly. These data suggest that rAAV-P2 treatment can lead to the opening of the BBB in the rat GBM model, most likely due to the pyroptosis-induced inflammation in the brain. Moreover, immunoglobulin G (IgG) staining assay was performed, which demonstrated that IgG invaded the brain through the BBB after rAAV-P2 treatment (Supplementary Fig. 8c).

As the long-term fragility of the BBB may lead to numerous brain diseases, the long-term integrity of the BBB was further investigated after rAAV-P2 treatment. Figure 3f–g showed a very low level of the Evans blue staining in the brain 43 days after rAAV-P2 treatment. Moreover, only a few IgG staining could be observed in the brain tissues (Supplementary Fig. 8c). These results suggest that the opening of the BBB caused by rAAV-P2 treatment is transient and can recover. Next, T cell infiltration was analyzed in the tumors by CD3-BV421 staining, which showed that the rAAV-P2 group had significantly higher CD3[+] T cell infiltration than that of the control groups (Fig. 3h and Supplementary Fig. 8d). Together, these data suggest that the pyroptosis induced by rAAV-P2 treatment can temporarily open the BBB, initiate T cells infiltration into the tumor, and subsequently eliminate the tumor cells.

The therapeutic effect of rAAV-P1 treatment was also investigated on the rat GBM model. As shown in Supplementary Fig. 8e–j, although rAAV-P1 treatment has limited effect on the progression of GBM, rAAV-P1 treatment caused striking cavitation in the brain tumor inoculation site (Supplementary

Fig. 8i, j). We speculated that this might be due to the low activity of the mCBA promoter, which can only drive a slow and minor expression of GSDMD[NT]. In this scenario, it cannot induce strong pyroptosis to repress the rapid tumor growth, and the overgrowth of the tumor can occupy a big volume in the brain. Although it can be eventually eliminated by rAAV-P1 treatment, the irreversible damage may lead to cavitation in the brain. Overall, there was only a very minor effect on the survival of the GBM rat model by rAAV-P1 treatment. Thus, in the long run, rAAV-P1 treatment may be suitable for some slowly progressing tumors.

**rAAV-P2 treatment inhibits breast cancer growth via activating antitumor immunity.** Next, the therapeutic effect of rAAV-P2 was evaluated on triple-negative breast cancer (TNBC). To this end, an orthotopic TNBC mouse model was established by injecting 4T1-luc cells expressing luciferase into the fat pad on the fourth left breast of Balb/c mouse. Six days post-tumor implantation, the mice were randomly assigned for intratumor injection with rAAV-P2 or rAAV-DIO-GSDMD[NT] (Fig. 4a). At 30 days post-tumor implantation, the average 4T1 tumor volume of the mice injected with rAAV-DIO-GSDMD[NT] exceeded over 1000 mm³, which was significantly larger than that of the rAAV-P2 treatment group (Fig. 4b, c). This result indicates that rAAV-P2 treatment can effectively inhibit the progression of TNBC, whereas the infection of rAAV-DIO-GSDMD[NT] cannot initiate the expression of GSDMD[NT] in the tumor cells. This is further supported by the solid tumor weight analysis of rAAV-P2- or rAAV-DIO-GSDMD[NT]-treated groups. Treatment with rAAV-P2 treatment significantly reduced the tumor weight when compared to the treatment with rAAV-DIO-GSDMD[NT] (Fig. 4d, e).

To investigate the impact of rAAV-P2 treatment on the TME, single-cell RNA-sequencing (scRNA-seq) analysis was performed on 4T1 tumor-infiltrating lymphocytes in mice treated with rAAV-DIO-GSDMD[NT] or with rAAV-P2 (Supplementary Fig. 9). The results of scRNA-seq analysis showed that rAAV-P2 treatment can increase the proportion of CD4[+], CD8[+] T cells, and NK cells in the tumor (Fig. 4f, g), suggesting an increased lymphocyte infiltration in 4T1 tumors. Consistent with this, our flow cytometry analysis of tumor-infiltrating lymphocytes showed that there were significantly more CD4[+], CD8[+] T cells, and NK cells in the tumors of the rAAV-P2 treatment group than those compared to the control group (Fig. 4h). This phenomenon is further confirmed by the immunostaining of CD3[+] T cells in the tumor tissue sections (Supplementary Fig. 12b–e). Together, these data suggested that rAAV-P2 treatment-induced tumor pyroptosis can recruit tumor-infiltrating lymphocytes.

The cell-to-cell communication in tumor-infiltrating lymphocytes plays a crucial role in the immune status in the TME. Thus, ligand–receptor analysis was performed based on the single-cell data, to compare cell-to-cell communication between the rAAV-

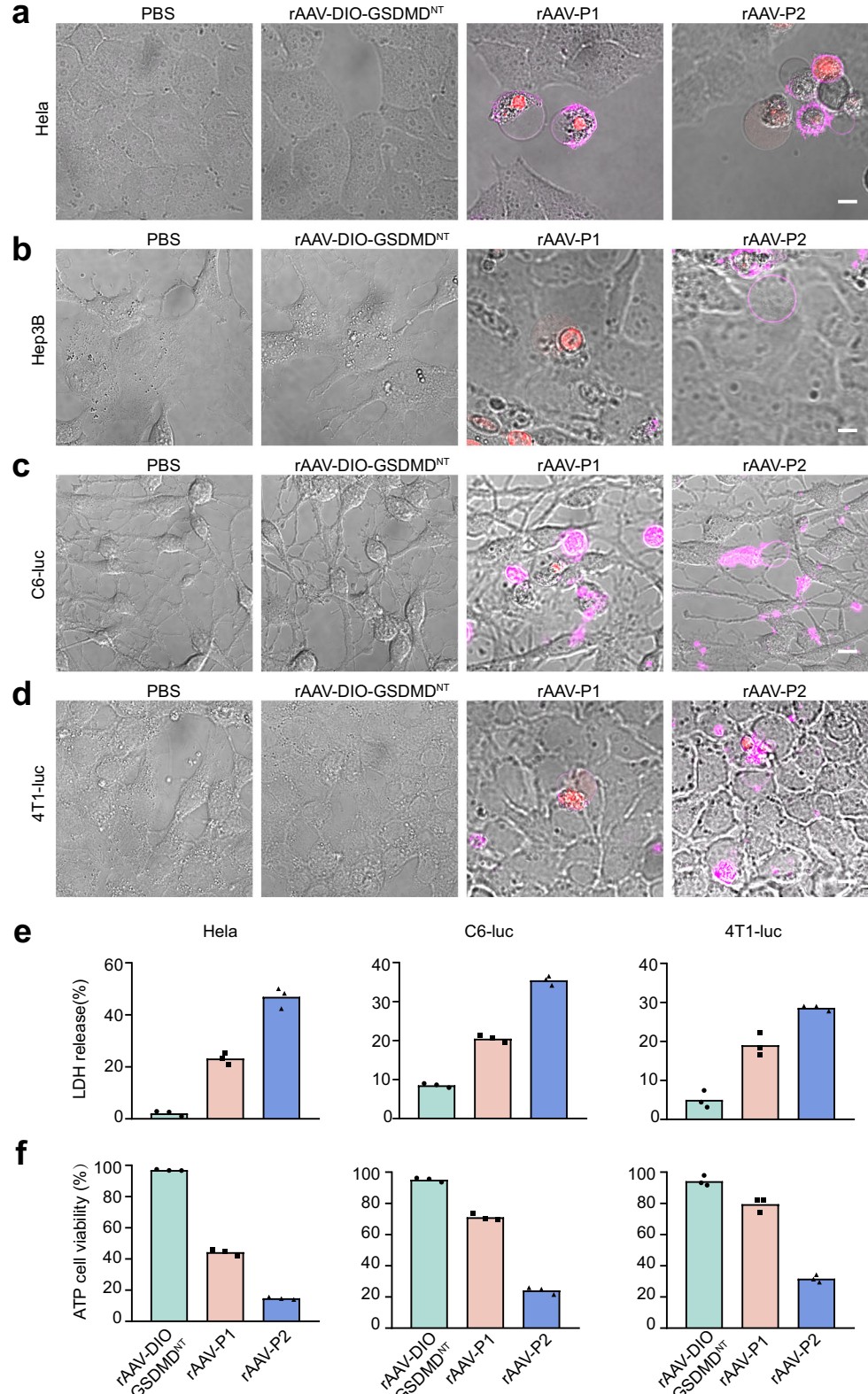

**Fig. 2 The pyroptosis induced by rAAV-P1 and rAAV-P2 in different cancer cell lines. a–d** Confocal images of the rAAV-treated HeLa (**a**), Hep3B (**b**), C6-luc (**c**), and 4T1-luc (**d**) cells. Scale bars, 20 μm. All the cells were added with propidium iodide and Annexin V-APC 15 min before imaging. **e**, **f** Comparison of LDH release-based cell death assay (**e**) and ATP cell viability assay (**f**) in HeLa, C6-luc, and 4T1-luc cells after treatment with rAAV-DIO-GSDMD$^{NT}$ (rAAV-ef1α-DIO-GSDMD$^{NT}$), rAAV-P1, and rAAV-P2, respectively. All data are representative of three independent experiments. Source data are provided as a Source Data file.

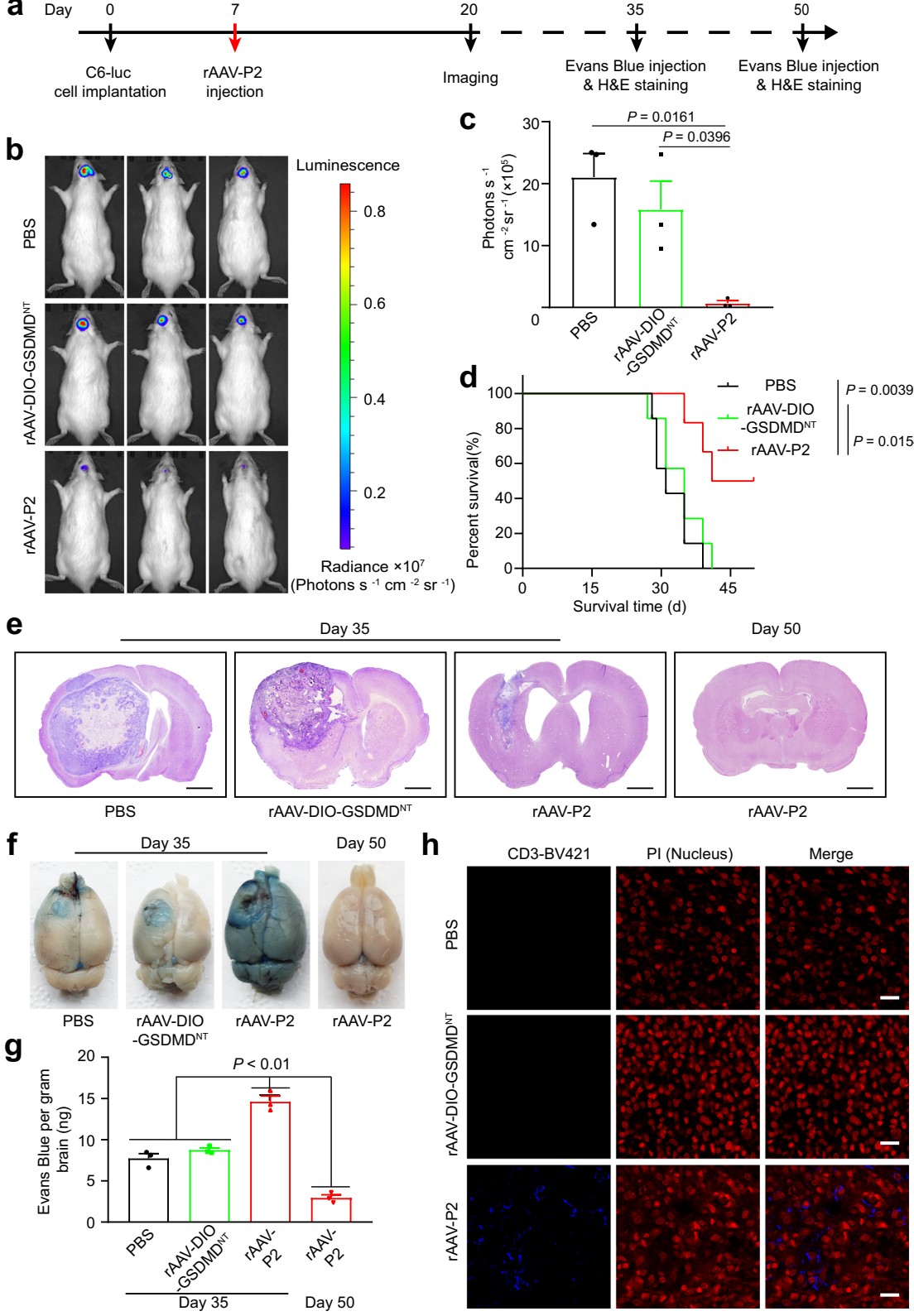

P2 treatment group and control group (Supplementary Fig. 10). Notably, the number of ligand–receptor interactions between the immunosuppressive cell population (MDSC) and other cell populations were decreased in the rAAV-P2 treatment group (Fig. 4i). Moreover, the interaction strength was weaker after treatment (Supplementary Fig. 10a). However, the cell-to-cell communication interaction number of the macrophage and

CD8[+] T cells to other cells was all increased (Supplementary Fig. 10). The expression analysis of the protumoral immunosuppressive genes and proinflammatory chemokines in each cell cluster further supported the conclusion that rAAV-P2 treatment can enhance the antitumor immunity in the TME (Supplementary Fig. 11). Of note, the athymic Nu/Nu mice TNBC model without mature T cells did not show tumor regression after

**Fig. 3 rAAV-P2 treatment prolongs the survival time of glioblastoma (GBM) rat model. a** Experimental timeline for rAAV-P2 treatment in GBM rat model. **b** Luciferase imaging of C6 tumors 20 days post implantation ($n = 3$ rats). **c** Corresponding quantification of luciferase expression in (**b**). Mean± s.e.m., $n = 3$ rats for each group, two-tailed unpaired $t$ test with Welch's correction. **d** Survival curves of C6-luc tumor-bearing rats treated with PBS, rAAV-DIO-GSDMD[NT] (rAAV-ef1α-DIO-GSDMD[NT]), and rAAV-P2, respectively, as indicated in (**a**). $n = 7$ rats for PBS and rAAV-DIO-GSDMD[NT] (rAAV-ef1α-DIO-GSDMD[NT]), six rats for rAAV-P2. Log-rank test was used for comparing two groups (PBS vs AAV-DIO-GSDMD[NT] or PBS vs rAAV-P2). **e** H&E-stained coronal sections of C6-luc tumor-bearing rat brains treated as indicated, 35 days (three on the left) and 50 days (right) post-tumor cell implantation, respectively. Representative images of each group ($n = 3$ rats) are presented. Scale bar, 2000 μm. **f** Evans Blue staining of whole brains treated as indicated, 35 days (three on the left) and 50 days (right) post-tumor cell implantation, respectively. Representative images of each group ($n = 3$ rats) are presented. **g** Corresponding quantification of Evans Blue in (**f**). Data were expressed as mean ± s.e.m., $n = 3$ rats for each group. Two-tailed unpaired Student's $t$ test was used for comparing the difference between two groups. PBS vs rAAV-P2 (day 35), $p = 0.0016$; rAAV-DIO-GSDMD[NT] vs rAAV-P2 (day 35), $p = 0.0014$; rAAV-P2 (day 50) vs rAAV-P2 (day 35), $p = 0.0001$. **h** Representative fluorescence images of CD3-BV421-stained of the C6-luc tumors following treatment as indicated. $n = 3$ rats for each group. Scale bar, 20 μm. All data are representative of three independent experiments. Source data are provided as a Source Data file.

rAAV-P2 treatment (Supplementary Fig. 13), suggesting that the therapeutic effect of rAAV-P2 mainly rely on the immune response induced by pyroptosis.

To further study the response of the whole 4T1 tumor tissue to rAAV-P2 treatment, the bulk RNA-seq analysis was performed on the whole tumor tissue treated with rAAV-DIO-GSDMD[NT] or rAAV-P2 (Supplementary Fig. 14). A total of 979 significantly differentially expressed genes (DEGs) were identified between these two groups (Supplementary Fig. 14a). To determine the biological functions of these DEGs, we performed Gene Ontology (GO) analysis. These DEGs mostly belong to genes involved in the immune-related signaling pathways, such as the lymphocyte proliferation pathway, innate immune response, and positive regulation of chemokine production pathways (Supplementary Fig. 14b). The DEGs in the positive regulation of the T cell proliferation signaling pathway were further analyzed by using STRING database[21] (Fig. 4j). It was found that the chemokines genes such as *Ccl5*, *Cxcl9*, and *Cxcl10* in the rAAV-P2 treatment group were significantly upregulated (Fig. 4j and Supplementary Fig. 14a). These data are consistent with findings reported by previous studies that CCL5 and CXCL9 can highly induce lymphocyte infiltration when they are both present in tumor tissues[22]. These data further validated the results of the scRNA-seq and flow cytometry analysis.

Moreover, our bulk RNA-seq analysis revealed that rAAV-P2 treatment can upregulate genes that are crucial for T cell activation (*Zap-70*, *Cd28*, *Cd86*, and *Cd3e*) and antitumor effects (*Gzmb*, *Gzmk*, and *Il12rb1*) (Fig. 4j and Supplementary Fig. 14a). Of note, *Il6*, which can promote tumor progression, was significantly downregulated (Fig. 4j and Supplementary Fig. 14a). The altered expression of these genes may activate antitumor immunity and inhibit TNBC progression. Importantly, we also observed the upregulation of the immune checkpoint genes such as *Cd274* (encoding PD-L1), *Pdcd1lg2* (encoding PD-L2), and *Pdcd1* (encoding PD1) in the rAAV-P2 treatment group (Fig. 4j and Supplementary Fig. 14a). These results suggest that the combination of rAAV-P2 treatment with immune checkpoint blocker such as anti-PD-L1 antibody or inhibitors might further improve the therapeutic effect of rAAV-P2 treatment on TNBC.

**Anti-PD-L1 therapy improves the oncolytic effect of rAAV-P2 treatment on TNBC in the mouse model.** To test whether anti-PD-L1 therapy can improve the oncolytic effect of rAAV-P2 treatment on TNBC in the mouse model, anti-PD-L1 therapy was performed 8 days post-rAAV-P2 intratumoral injection in TNBC (Fig. 5a). This timepoint was chosen because there is a time window for rAAV-mediated gene expression[23,24]. The luciferase live imaging showed that rAAV-P2 combined with anti-PD-L1 therapy indeed is significantly more effective than rAAV-P2 treatment alone

in reducing the sizes of TNBC tumors (Fig. 5b, c). Notably, the treatment with anti-PD-L1 alone had limited effect in preventing tumor growth (Fig. 5b–g), probably due to the immunosuppression caused by TME. This hypothesis was confirmed by the results of flow cytometry analysis (Supplementary Fig. 15). Compared with the control group and the anti-PD-L1 therapy-only group, there were significantly more propidium iodide-positive cells in the tumors in the rAAV-P2 and anti-PD-L1 combination therapy group (Fig. 5h). In addition, obvious cavities formed by clusters of propidium iodide-positive cells were observed in the tumors in the rAAV-P2 and anti-PD-L1 combination therapy group (Fig. 5h), suggesting that the combined anti-PD-L1 therapy can further improve the effect of rAAV-P2 treatment. This effect is also supported by the tumor weight and volume analyses (Fig. 5d–g). rAAV-P2 treatment combined with anti-PD-L1 therapy can eventually eliminate almost the entire tumor tissue.

Furthermore, we developed a packaging strategy to express GSDMD[NT] in another clinically applied oncolytic virus vector, adenovirus vector. As the adenoviral vector AdMax packaging system contains Cre recombinase already, we selected Flp recombinase for this packaging strategy to avoid the expression of GSDMD[NT] (Supplementary Fig. 16a). Ad5-Flp and Ad5-fDIO-GSDMD[NT] were packaged (Supplementary Fig. 16b) and mixed (the mixture called Ad5-P2), and then tested on HEK 293, HeLa, and 4T1-luc cells, respectively. As shown in Supplementary Fig. 16c, d, Ad5-P2 can induce pyroptosis in HEK 293, HeLa, and 4T1-luc. This result indicates that this packaging strategy can be successfully applied for the expression of GSDMD[NT] in adenovirus vectors (Supplementary Fig. 17). The therapeutic effect of Ad5-P2 was then evaluated in an orthotopic TNBC mouse model. Six and 14 days post-tumor implantation, PBS, Ad5-P2, and rAAV-P2 were injected into the tumor of the mouse model, respectively (Supplementary Fig. 18a). Our data showed that Ad5-P2 treatment inhibited tumor growth in the TNBC mice model (Supplementary Fig. 18), suggesting that Ad5-P2-mediated pyroptosis may be also applied for tumor treatment.

## Discussion

GSDM[NT] can induce pyroptosis and subsequently initiate a strong inflammatory response that holds a great potential to activate robust antitumor immunity[4–6]. However, it is challenging to efficiently produce and deliver GSDM[NT] into cancer cells due to its high intracellular toxicity to almost all mammalian cells and even to bacteria. A previous study proposed a bio-orthogonal system using nanomaterials to deliver GSDMA3(N + C) into tumors and stimulate antitumor immunity for oncolysis[5]. Although this system could be potentially used to induce pyroptosis and applied for antitumor therapy, the nanomaterials have some disadvantages such as the physiological interaction and metabolism for clinical practice, the side effect, and the

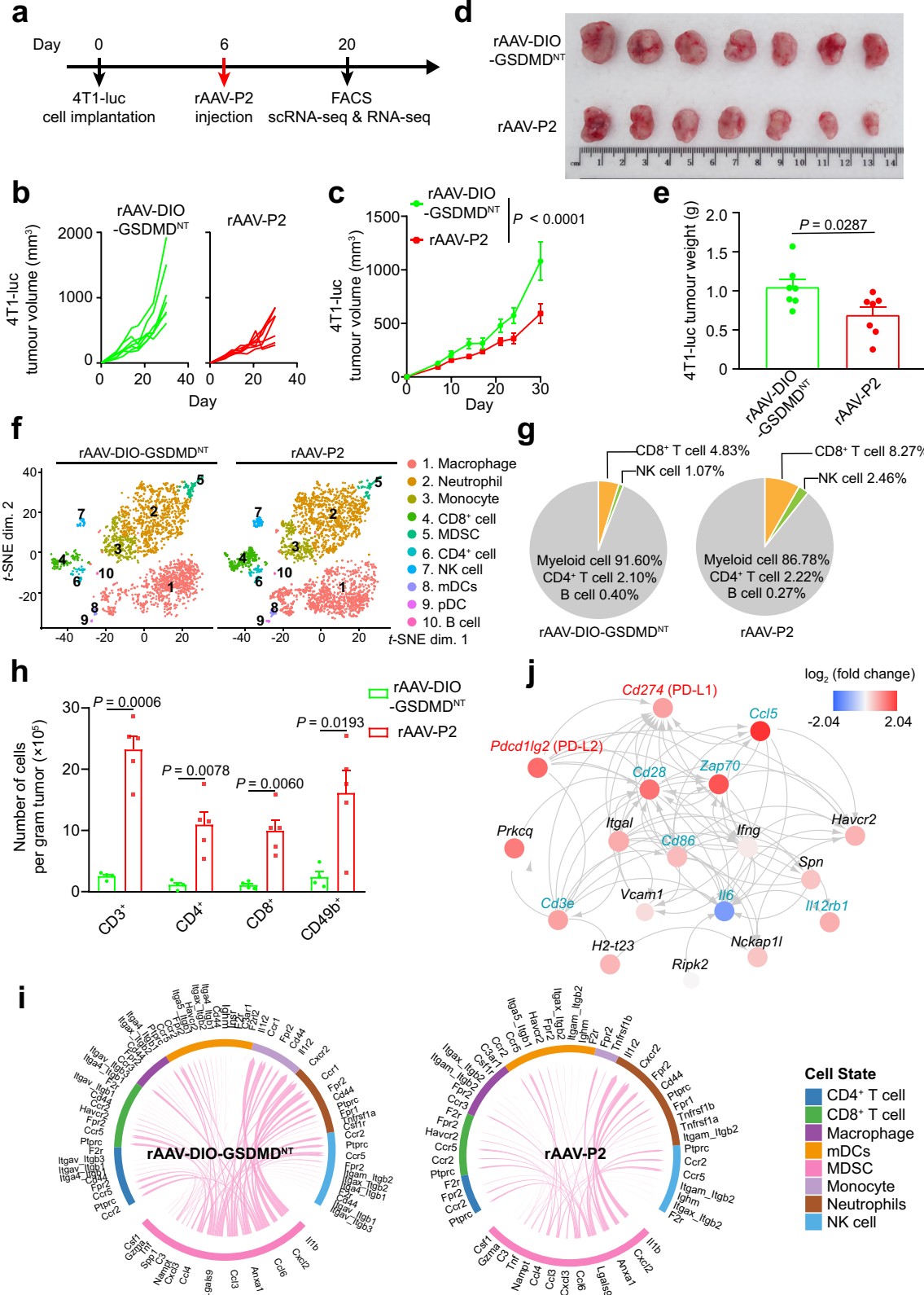

cost[25]. Applying replication-defective oncolytic viruses, such as rAAV, to deliver $GSDM^{NT}$ into tumors may be an alternative promising strategy. However, owing to the extremely high cytotoxicity of $GSDM^{NT}$ during packaging, the conventional rAAV packaging approaches are impossible to achieve rAAV-$GSDM^{NT}$ with enough titer for clinical use. A Schwann-cell specific promoter has been employed to control the expression of $GSDM^{NT}$

during AAV packaging, which might be specifically used for schwannomas treatment[18]. However, different cell-specific promoters have different levels of leakage expression, which may affect the titer of the virus and increase the cost for the clinical gene therapy. In this study, we developed two strategies and successfully packaged rAAV-$GSDM^{NT}$, which can be potentially applied for antitumor therapy. The compatibility between

**Fig. 4 rAAV-P2 treatment activates antitumor immunity and inhibits the growth of triple-negative breast cancer (TNBC). a** Schematic of rAAV-P2 treatment on TNBC mouse model. **b** Tumor volume of an individual mouse. $n = 7$ mice per group. **c** Average tumor volume of mice as indicated. $n = 7$ mice per group. Data were expressed as mean ± s.e.m. Two-way ANOVA with the Geisser–Greenhouse correction was used for comparing two groups. **d** Photographs of the tumors 30 days post treatment with rAAV-P2 and rAAV-DIO-GSDMD$^{NT}$ (rAAV-ef1α-DIO-GSDMD$^{NT}$), respectively. **e** Average tumor weight of mice as indicated. $n = 7$ mice per group. Data were expressed as mean ± s.e.m. Two-tailed unpaired Student's $t$ test was used for comparing two groups. **f** t-SNE plots of tumor-infiltrating single CD45$^+$ immune cells of 4T1 tumors from mice treated with rAAV-DIO-GSDMD$^{NT}$ ($n = 2$ mice) and rAAV-P2 ($n = 2$ mice), respectively. **g** The relative frequencies of different clusters. **h** Quantification of the tumor-infiltrating lymphocytes from TNBC mouse model. $n = 4$ mice for rAAV-DIO-GSDMD$^{NT}$ (rAAV-ef1α-DIO-GSDMD$^{NT}$) and 5 mice for rAAV-P2. Data were expressed as mean ± s.e.m. Two-tailed unpaired $t$ test with Welch's correction was used for comparing two groups. **i** Condition-specific linkages between MDSC ligands and other lymphocyte cluster receptors in the rAAV-DIO-GSDMD$^{NT}$ and rAAV-P2 treatment groups. **j** Diagram of the DEG interaction network of "the positive regulation of T cell proliferation" signaling pathway based on the STRING database. Data shown (**a–e**, **h**) are representative of two independent experiments. Source data are provided as a Source Data file.

transcription factor and the underlying promoter is pivotal for gene expression. To maximize the divergence between the promoter and the potential transcription factors we employed a mammalian promoter and packaged the virus in insect cells, as the insect cell AAV packaging system is well established for large-scale AAV production by fermentation. After screening for several promoters, our data suggested that sf9 insect cells do not express the transcription factors to drive mCBA promoter activity (a promoter that can initiate gene expression in most mammalian cells). Thus, this strategy can produce high titers of rAAV-GSDM$^{NT}$ in sf9 cell that can be used for different types of tumor therapy. By using a mammal-specific promoter to drive GSDMD$^{NT}$ expression and package the virus in Sf9 insect cell system, we can significantly reduce the toxicity during virus preparation.

In our second strategy, the double-floxed GSDMD$^{NT}$ was invertedly cloned into an rAAV vector, which can be reverted by the coinfection of rAAV-Cre during treatment. In this way, it can completely avoid the expression of GSDMD$^{NT}$. By these approaches, we obtained rAAV-P1 and rAAV-P2 viruses with high titer to induce pyroptosis for tumor therapy. Considering the low activity of the mCBA promoter, rAAV-P1 may be suitable for the treatment of some slow-progressing tumors. While rAAV-P2 was packaged by HEK 293T cells in this study, it can also be packaged through the Sf9 cell system on a large scale. Other than rAAV-GSDM$^{NT}$ packaging, our strategies can also be extensively used to express and deliver other toxic proteins, such as thymidine kinase 1[26] and trichosanthin[27], which could also be potentially applied for tumor therapy.

The existence of the BBB obstructs the conventional chemotherapy drugs and immune cells for intracranial tumors[20,28], which severely hampers the progress of clinical treatments for GBM. Our study proves that rAAV-P2-induced pyroptosis can temporarily open the BBB and increase T cell infiltration in GBM tumors, and thereby significantly prolonging the survival time in the rat GBM model. These data suggest that rAAV-P2 has a promising potential for tumor treatment. Notably, our research also proposes that during the process of GBM rAAVs therapy, extra anti-infection actions need to be taken to avoid infection due to the temporary opening of the BBB. As rAAV-P2 treatment can temporarily open the BBB, it may be potentially applied to the therapy of other brain diseases to facilitate the oversize drugs to permeabilize the BBB.

Immune checkpoint blockade therapy such as anti-PD1/PD-L1 has been successfully applied in clinical treatments[29,30]. However, this approach is only effective in certain types of cancers, probably due to the distinct immunosuppressive TME[31,32]. Our RNA-seq and tumor-infiltrating lymphocyte analysis data show that rAAV-P2 treatment can upregulate the expression of chemokine and cytokine-related genes and increase lymphocyte infiltration in the tumors. Consistent with previous studies[4–6], our data reveal that the

pyroptosis induced by GSDM$^{NT}$ can alter the immunosuppressive TME. In addition, the upregulation of the immune checkpoint (PD1/PD-L1) genes implies that anti-PD-L1 therapy can enhance the oncolytic effect of rAAV-GSDMD$^{NT}$. Accordingly, our data demonstrated that the combination of anti-PD-L1 therapy and rAAV-P2 can indeed improve the oncolytic effect of rAAV-P2 and have a promising prospect in antitumor immunotherapy.

In our study, the rAAV-GSDM$^{NT}$ viruses were directly injected into the tumors to achieve tumor targeting. In this scenario, the application of rAAV-GSDM$^{NT}$ by this approach may be limited to the treatment of solid tumors. It has been demonstrated that rAAV targeting can be achieved by incorporating high-affinity ligands into the viral vector particles, such as Her2-rAAV, to realize specific gene transference into Her2/neu-positive tumor[33]. Furthermore, the progress in tumor-specific promoters has also provided ample alternative solutions for the GSDM$^{NT}$ tumor targeting[34]. Collectedly, our study developed two strategies to circumvent the cytotoxicity of GSDMD$^{NT}$ and efficiently produce and deliver GSDM$^{NT}$ into tumor cells. It may pave the way for the oncolytic therapy of cancer via pyroptosis.

## Methods

**Animal ethics**. All animal studies were performed in accordance with the Guide for the Care and Use of Laboratory Animals of the Research Ethics Committee of Huazhong Agricultural University. The use of mice and rats has been approved by the Research Ethics Committee of Huazhong Agricultural University, Hubei, China (HZAUMO-2019-041). All animals were kept in a specific pathogen-free environment with a dark/light cycle of 12/12 h (daily light time, 8:00–20:00), a temperature of 18–29 °C, and relative humidity of 40–70%. In the C6-luc GBM rat model, the prescribed treatment was performed 8 days post-C6-luc cell inoculation. When the continuous body weight decreased by over 20%, the rats were anesthetized and sacrificed. In the 4T1-luc breast cancer model, the prescribed treatments were performed when the tumor volume reached 100 mm$^3$ (about 7 days post inoculation). In all experiments, none of the mice had tumors larger than 2000 mm$^3$. No severe abdominal distension exceeded 10% of its original body weight was observed.

**Plasmid, antibody, and drug**. The complementary DNA (cDNA) encoding human GSDMD and GSDME were a gift from Professor Jiahuai Han (Xiamen University). The cDNA encoding human GSDMB was obtained from ABclonal Technology Company. The mammal-specific CBA promoter was provided by Professor Jingping Ge (Heilongjiang University). The pFastBac-ITR-CMV-eGFP and pCAGGS plasmids were provided by Professor Guiqing Peng (Huazhong Agricultural University). The three rAAV plasmids (pAAV-CMV-eGFP, pAAV-DJ, and pHelper), pAAV-ef1α-DIO, pAAV-Cre, pHKO-luc, pSPAX, pMD2.G, pFlp, and pfDIO were provided by Professor Fuqiang Xu (Wuhan Institute of Physics and Mathematics, Chinese Academy of Sciences). The pHBad-eGFP, pBHGloxΔE1/3 Cre plasmids were provided by Professor Anding Zhang (Huazhong Agricultural University). The CMV promoter in pFastBac-ITR-CMV-eGFP was replaced with mCBA promoter, CBA promoter (amplified from pCAGGS), and CAG promoter (amplified from pCAGGS) sequence to obtain pFastBac-mCBA-eGFP, pFastBac-CBA-eGFP, and pFastBac-CAG-eGFP, respectively. The GSDMD$^{NT}$ was amplified from the human cDNA and used to replace the eGFP gene in pAAV-CMV-eGFP, pFastBac-ITR-CMV-eGFP, pFastBac-mCBA-eGFP, and pFastBac-CAG-eGFP to obtain pAAV-CMV-GSDMD$^{NT}$, pFastBac-CMV-GSDMD$^{NT}$, pFastBac-mCBA-GSDMD$^{NT}$, and pFastBac-CAG-GSDMD$^{NT}$, respectively. pAAV-ef1α-DIO-GSDMD$^{NT}$, pAAV-ef1α-DIO-GSDMB$^{NT}$, and

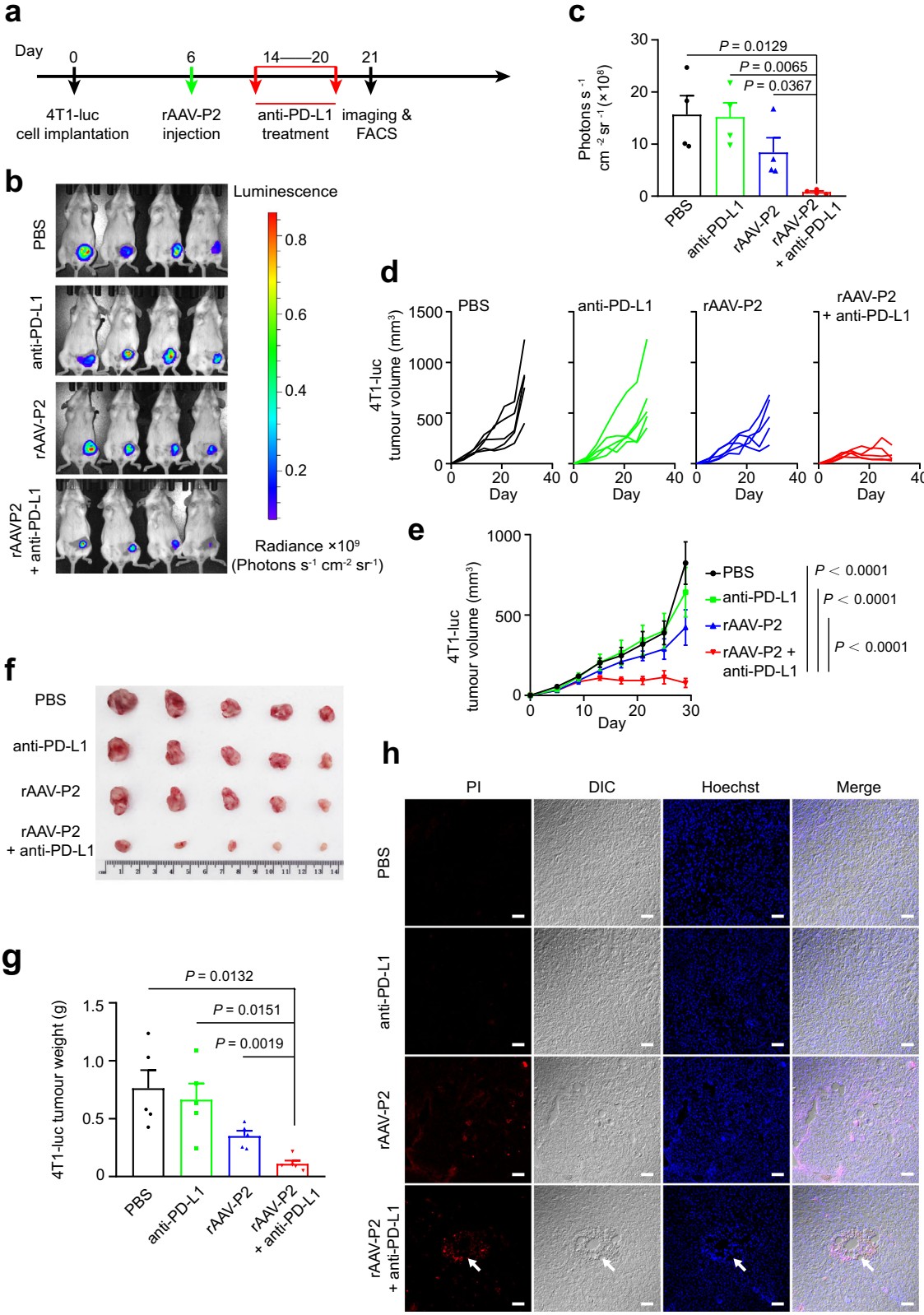

pAAV-ef1α-DIO-GSDME^NT were obtained by inserting *GSDMD^NT*, *GSDMB^NT*, and *GSDME^NT* into pAAV-ef1α-DIO. The ef1α promoter in pAAV-ef1α-DIO-GSDMD^NT was replaced by mCBA promoter sequence to obtain pAAV-mCBA-DIO-GSDMD^NT. The *luc* was amplified from pHKO-luc and used to replace the eGFP gene in pAAV-CMV-eGFP to obtain pAAV-CMV-luc. The *fDIO* and *Flp* were amplified from pFlp and pfDIO, respectively. The *fDIO-GSDMD^NT* was obtained by *fDIO* and *GSDMD^NT* through overlap extension PCR. pHBad-fDIO-

GSDMD^NT and pHBad-Flp were obtained by inserting *fDIO-GSDMD^NT* and *Flp* into pAAV-ef1α-DIO.

Anti-rat antibody CD3-BV421 (clone 1F4) was purchased from BD Biosciences (USA). Anti-rat IgG-488 was purchased from Invitrogen (USA). Antibodies for flow cytometry analysis, anti-mouse CD16/32 (clone 93), CD3-PE (clone 17A2), CD4-FITC (clone RM4.5), CD8a-PacBlue (clone 53-6.7), CD49b-APC (clone DX5), and CD45-APC-Cy7 (clone 30-F11) were purchased from BioLegend (USA).

**Fig. 5 Anti-PD-L1 therapy boosts the effect of rAAV-P2 treatment on TNBC in the mouse model. a** Schematic of rAAV-P2 combined with anti-PD-L1 treatment on TNBC mouse model. **b** Luciferase imaging of 4T1-luc breast tumors 21 days post-tumor implantation. **c** Corresponding quantification of luciferase expression in (**b**). Mean ± s.e.m., from four independent replicates, one-tailed unpaired $t$ test with Welch's correction. **d** Tumor volume of sn individual mouse. $n = 5$ mice per group. **e, g** Average tumor volume (**e**) and weight (**g**) of mice as indicated. $n = 5$ mice per group. Data were expressed as mean ± s.e.m. Two-way ANOVA with the Geisser–Greenhouse correction was used for comparing the difference between two groups (**e**). Two-tailed unpaired $t$ test with Welch's correction was used for comparing the difference between two groups (**g**). **f** Photographs of representative tumors 29 days post treatment. **h** Propidium iodide staining of tumor cell pyroptosis induced by treatment as indicated. $n = 3$ mice for each group. Scale bar, 20 μm. All data are representative of two independent experiments. Source data are provided as a Source Data file.

Anti-PD-L1 blocker BMS202 (cat. no. HY-19745) was purchased from MedChemExpress (USA).

**Cell lines**. HeLa and HEK 293T cells were obtained from the American Type Culture Collection. Hep3B and Sf9 cells were gifted by Professor Mingqian Feng (Huazhong Agricultural University) and Professor Guiqing Peng, respectively. 4T1 and C6 cells were purchased from the Cell Resource Center of Shanghai Institutes for Biological Sciences, Chinese Academy of Sciences.

To construct 4T1-luc and C6-luc cell lines, plasmids pHKO-luc, pSPAX and pMD2.G were cotransfected into HEK 293T with a ratio of 1:1:1 for 60 h. The supernatant was collected and concentrated with lentivirus concentrate to obtain lentivirus-luc. Lentivirus-luc was used to infect 4T1 and C6 cells, and single positive cells were picked and proliferated to obtain 4T1-luc and C6-luc cell lines expressing luciferase.

HeLa, HEK 293T, and Hep3B cells were cultured in Dulbecco's modified Eagle's medium containing 10% (v/v) fetal bovine serum (FBS) and 2 mM L-glutathione. 4T1 and 4T1-luc cells were grown in RIPM1640 medium containing 10% FBS, 1 mM sodium pyruvate (Gibco), 1× GlutaMAX$^{TM}$-1 (Gibco), and 1× MEM NEAA (Gibco). C6 and C6-luc cells were cultured in Ham's F-12K (Kaighn's) medium containing 10% FBS. All the cells were grown at 37 °C in a 5% $CO_2$ incubator. Myco-Blue Mycoplasma Detector Kit (Vazyme) was used to detect mycoplasma contamination in each culture once every 3 months. All the cells were identified by short tandem repeat analysis.

The *Escherichia coli* DH10Bac-competent cells were produced in our lab following the standard protocol for competent cell generation.

**Viruses**. Viral vectors were constructed in accordance with the Biosafety Guidelines of the Huazhong Agricultural University Administrative Biosafety Committee on Laboratory. Four baculoviruses were used in the current study, including rBac-AAV/helper, rBac-mCBA-eGFP, rBac-CMV-GSDMD$^{NT}$, and rBac-mCBA-GSDMD$^{NT}$. rBac-AAV/helper was provided by Professor Fuqiang Xu. rBac-CMV-eGFP, rBac-mCBA-eGFP, rBac-CMV-GSDMD$^{NT}$, and rBac-mCBA-GSDMD$^{NT}$ were constructed according to the Bac-to-Bac Expression Kit Handbook (Invitrogen).

*rAAV-P1 packaging*. As shown in Supplementary Fig. 3, pFastBac-ITR-mCBA-GSDMD$^{NT}$ was transformed into DH10Bac-competent cells and plated on an LB agar plate containing 50 μg/ml kanamycin, 7 μg/ml gentamicin, 10 μg/ml tetracycline, 100 μg/ml Blu-gal, and 40 μg/ml IPTG (blue-white spot screening LB agar plate). After 48 h incubation at 37 °C, the white spot colonies were picked and streaked on the blue-white spot screening LB agar plate and incubated at 37 °C for 12 h. Subsequently, the white colonies were picked for colony PCR identification. The positive colonies were cultured in LB liquid medium containing 50 μg/ml kanamycin, 7 μg/ml gentamicin, and 10 μg/ml tetracycline. The Bacmid DNA was extracted using the PureLink$^{TM}$HiPure Plasmid DNA Miniprep Kit (Thermo Fisher). The transfection mixture was added to preseeded Sf9 cells and cultured in a biochemical incubator at 27 °C for 4–5 days, of which the supernatant was harvested as rBac-mCBA-GSDMD$^{NT}$ P1 viral stock. The P1 viral stock was further infected into Sf9 cells at a multiplicity of infection (MOI) = 0.1. The infected Sf9 cells were further cultured, of which the supernatant was collected to obtain high-titer rBac-mCBA-GSDMD$^{NT}$ P2 viral stock. Plaque assay was used to measure the viral stock to ensure that the rBac-mCBA-GSDMD$^{NT}$ and rBac-AAV/helper titers were >1 × 10$^8$ PFU/ml.

Sf9 cells were infected with rBac-AAV/helper and rBac-mCBA-GSDMD$^{NT}$ infected at MOI = 1, respectively. The cells density was kept at 2 × 10$^6$ cells/ml. The cells were cultured in a shaking incubator at 27 °C for 72 h and collected by centrifugation at 500 × $g$ for 5 min. The cells were resuspended in PBS at a density of 2 × 10$^7$ Sf9 cells per ml and then frozen and thawed four times with liquid nitrogen and a 37 °C water bath to disrupt the cells. The supernatant was centrifuged at 10,000 × $g$ for 10 min and bathed at 60°C for 30 min to inactivate the residual baculoviruses. Fifty units per ml DNase I was added and digested at 37 °C for 45 min to digest the remaining genomic DNA. After the digestion, the mixture was centrifuged at 10,000 × $g$ at 4 °C for 10 min. Then, the supernatant was purified from the concentrated mixture using Iodixanol Gradient Ultracentrifugation method (Addgene). After purification, a 100 kDa protein ultrafiltration tube was applied to obtain rAAV-P1 (rAAV-mCBA-GSDMD$^{NT}$). The rBac-CMV-eGFP, rBac-CMV-GSDMD$^{NT}$, and rBac-mCBA-eGFP were used to package rAAV/Sf9-CMV-eGFP, rAAV-CMV-GSDMD$^{NT}$, and rAAV-mCBA-eGFP using the same method.

*rAAV-P2 packaging*. As shown in Supplementary Fig. 4, the rAAV-P2 was packaged using the three plasmids packaging system protocol (Cell Biolabs). Taking rAAV-ef1α-DIO-GSDMD$^{NT}$ as an example, all the vectors (pAAV-ef1α-DIO-GSDMD$^{NT}$, pAAV-DJ, and pHelper) were mixed with the same ratio and cotransfected into HEK 293T cells. At 72 h post transfection, the transfected cells were collected by centrifugation at 500 × $g$ for 5 min and then resuspended in PBS at a density of 2 × 10$^7$ cells per ml. The cells were lysed by repeatedly freezing and thawing using liquid nitrogen and a 37 °C water bath. The lysed cells were centrifuged at 10,000 × $g$ for 10 min and the supernatant was collected. The supernatant was exposed to 50 U per ml DNase I at 37 °C for 45 min to digest the remaining genomic DNA. After digestion, the samples were centrifuged at 10,000 × $g$ for 10 min at 4 °C. Then, the virus (rAAV-ef1α-DIO-GSDMD$^{NT}$) was purified from the concentrated supernatant using Iodixanol Gradient Ultracentrifugation method (Addgene). rAAV-Cre, rAAV-CMV-luc, rAAV/293-CMV-eGFP, rAAV-ef1α-DIO-GSDMB$^{NT}$, rAAV-ef1α-DIO-GSDME$^{NT}$, and rAAV-CMV-GSDMD$^{NT}$ were obtained following the same method.

*rAAV titer measurement*. All the rAAVs were titrated by fluorescent quantitative PCR using ITR-F: 5′-CGGCCTCBATGAGCGA-3′ and ITR-R: 5′-GGAACCC-TAGTGATGGAGTT-3′ as amplification primers. pAAV-CMV-eGFP was used as a standard product according to Springer AAV Protocol. All rAAV titers were >1 × 10$^{12}$ vg/ml. The titer of rAAV-mCBA-GSDMD$^{NT}$ (rAAV-P1) was adjusted to 1 × 10$^{12}$ vg/ml. The titers of rAAV-Cre and rAAV-ef1α-DIO-GSDMD$^{NT}$ (rAAV-P2) were adjusted to 2 × 10$^{12}$ vg/ml and mixed (the final titers of both rAAVs were 1 × 10$^{12}$ vg/ml).

The Ad5-Flp and Ad5-fDIO-GSDMD$^{NT}$ were constructed based on the AdMax system (Supplementary Fig. 17). The infectious units of Ad5-Flp and Ad5-fDIO-GSDMD$^{NT}$ were titrated on HEK 293 cells using an AdenoX$^{TM}$ Rapid Titre Kit (Clontech, USA). The infectious units of Ad5-Flp and Ad5-fDIO-GSDMD$^{NT}$ (Ad5-P2) were adjusted to 2 × 10$^{10}$ PFU/ml and mixed for infection (the final titers of both Ad5 were 1 × 10$^{10}$ PFU/ml).

**Sequence alignment of promoters and evaluation of promoter activity**
*Promoter sequence alignment*. The mCBA promoter, CBA promoter, and CAG promoter sequences were saved in FASTA format and uploaded to the ClustalW website (https://www.genome.jp/tools-bin/clustalw) for sequence alignment using Clustal algorithm. Then "clusterw.aln" file was uploaded to the ENDscript/ESPript website (http://espript.ibcp.fr/ESPript/cgi-bin/ESPript.cgi) to map the results of the sequence alignment.

*Evaluation of mCBA promoter activity in insect Sf9 and mammalian cells*. pFastBac-ITR-CMV-eGFP, pFastBac-mCBA-eGFP, and pFastBac-CBA-eGFP were transfected into Sf9 cell lines preseeded in a 6-well cell culture plate, respectively. The concentration of each plasmid DNA was 1 μg plasmid per well. At 48 h post-transfection incubation, the cells were examined under the Invitrogen$^{TM}$ EVOS$^{TM}$ FL Auto 1 microscope. Sf9 cells were infected with rBac-CMV-GSDMD$^{NT}$, rBac-mCBA-eGFP, and rBac-mCBA-GSDMD$^{NT}$ at an MOI = 1, respectively. At 48 h post infection, the images were recorded on a ×40 objective with an Invitrogen$^{TM}$ EVOS$^{TM}$ FL Auto 1 microscope.

pFastBac-mCBA-eGFP, pFastBac-CAG-eGFP, and pFastBac-CBA-eGFP were transfected into HEK 293T, HeLa, and 4T1-luc cell lines preseeded in CellCarrier-96 ultra plates (PerkinElmer), respectively. At 48 h post transfection, images were collected on a ×20 objective with the Opera Phenix$^{TM}$ High Content Screening System. The average fluorescence intensity was analyzed with the Opera Phenix$^{TM}$ High Content Screening System.

For Cre/lox system plasmids pAAV-CMV-GSDMD$^{NT}$, pAAV-ef1α-DIO-GSDMD$^{NT}$, and pAAV-Cre transfected into preseeded HEK 293T cells in a 6-well plate at a concentration of 1 μg plasmid per well, respectively. Invitrogen$^{TM}$ EVOS$^{TM}$ FL Auto 1 microscope was used to record images on a ×40 objective 48 h post transfection.

**The rAAV-P1 and rAAV-P2 induced pyroptosis efficiency analysis**. In order to study the efficiency of pyroptosis mediated by ef1α and mCBA promoters, equimolar amounts of pAAV-ef1α-DIO-GSDMD$^{NT}$ and pAAV-mCBA-DIO-GSDMD$^{NT}$ were cotransfected into HEK 293T, HeLa, and 4T1-luc cell lines preseeded in 96-well plates with pAAV-Cre, respectively. At 48 h post transfection, images were collected

by the Olympus FV1000 microscope on a ×20 objective. The LDH-Glo Cytotoxicity Assay (Promega) was used to determine the LDH release assays.

In order to study whether the rAAV packaged by HEK 293T cells or Sf9 insect cells could cause different gene transduction efficiency, rAAV/293-CMV-eGFP and rAAV/Sf9-CMV-eGFP were infected into HEK 293T, HeLa, and 4T1-luc cell lines preseeded into CellCarrier-96 ultra plates (PerkinElmer), respectively. At 48 h post infection, images were collected by the Opera Phenix™ High Content Screening System on a ×20 objective. The average fluorescence intensity was analyzed with Opera Phenix™ High Content Screening System.

**Pyroptosis assays**. For pyroptosis assay in HEK 293T, the precultured HEK 293T cells were seeded into CellCarrier-96 ultra plates (PerkinElmer) and then infected with rAAV-ef1α-DIO-GSDMD$^{NT}$, rAAV-P1, and rAAV-P2, respectively. The cells treated with PBS were taken as a control group. All the cells were incubated at 37 °C, 5% $CO_2$ for 6 h and then examined with an Opera Phenix™ High Content Screening System (PerkinElmer). The images were collected every 30 min under a ×63 water objective. Then, an Olympus IX83 microscope with a 40-fold differential interference contrast objective was used post 48 h infection to further examine the pyroptosis in each group. All the image data shown represented at least three fields selected randomly.

HeLa, HEK 293T, Hep3B, 4T1-luc, and C6-luc cells were seeded on CellCarrier-96 ultra plates (PerkinElmer), respectively. The cells were inoculated with rAAV-DIO-GSDMD$^{NT}$, rAAV-P1, and rAAV-P2, respectively. The cells were washed by PBS at 48 h post infection and then stained with propidium iodide and annexin V-APC to detect cell morphology. Confocal images were collected on a ×40 objective using an Olympus FV1000 microscope. CellTiter-Glo Luminescent Cell Viability Assay (Promega) was used to determine cell viability. The LDH-Glo Cytotoxicity Assay (Promega) was used to determine the LDH release assays.

HeLa, HEK 293T, and 4T1-luc cells were seeded on 96-well plates, respectively. The cells were inoculated with rAAV-ef1α-DIO-GSDMD$^{NT}$, rAAV-ef1α-DIO-GSDMB$^{NT}$, rAAV-ef1α-DIO-GSDME$^{NT}$, rAAV-P2, rAAV-P2B, rAAV-P2E, and Ad5-P2, respectively. At 48 h post infection, images were collected on a ×20 objective using an Olympus FV1000 microscope. The LDH-Glo Cytotoxicity Assay (Promega) was used to determine the LDH release assays.

**C6-luc GBM rat model and oncolytic treatment**. All rats were purchased from the Experimental Animal Center of Huazhong Agricultural University. In order to construct a GBM rat model, C6-luc cells ($1 \times 10^6$) in 10 µl of F-12K medium were stereotactically injected into the left striatum of Wistar male rats (180–220 g)[35].

To verify the oncolytic effect of rAAV-P1, 7 days post-C6 cell implantation, the animals were randomly divided into two groups injected with $1 \times 10^{10}$ vg of rAAV-P1 or rAAV-mCBA-eGFP (Supplementary Fig. 4e). All the rats were kept in a standard environmental condition and the weight was recorded regularly. At 23 days post-rAAV treatment, the animals were sacrificed and the brains were harvested for the histopathology examination.

In order to check the antitumor effects of rAAV-P2, 7 days post-C6 cells implantation, the animals were randomly divided into three groups intratumorally injected with $1 \times 10^{10}$ vg of rAAV-P2, rAAV-DIO-GSDMD$^{NT}$, and PBS, respectively (Fig. 3a). The body weights were checked once every 2 or 3 days. A 75 mg/kg dose of D-luciferin (Promega) was intraperitoneally injected into three rats in each group 13 days post treatment. At 15 min post-D-luciferin injection, the live animal imagings were performed by using the Small Animal In Vivo Imager (PerkinElmer), and the imaging software was used to quantify the expression of luciferase. At 28 or 43 days post-rAAV treatment, the rats were anesthetized and sacrificed to harvest the brains for histopathology assays. In addition, the brains harvested 28 days post-rAAV treatment were frozen, sectioned, and stained with CD3-BV421 (1:200). The ImageJ software was used to quantitatively analyze the gray value of the collected images (three rats in each group and three random areas per rat) to evaluate the lymphocyte infiltration in the tumor.

**BBB integrity assessment**. At 28 or 43 days post-rAAV treatment, Evans Blue solution (2% in PBS) was injected into the viable tail vein at a dose of 1 ml per rat. At 1 h post injection, the rats were perfused with PBS to harvest the brains. After being photographed, the brain tissues were transferred into prelabeled tubes separately and then homogenized. After that, formamide (1 ml per 100 mg brain tissue) was added to each homogenized sample and incubated at 60 °C for 24 h and then centrifuged at $10,000 \times g$ for 10 min to collect the supernatant and recorded the absorbance at 620 nm by a spectrophotometer. The non-homogenized brain tissues from each group of animals were sectioned and stained with anti-rat IgG-488 (1:200). Confocal images of the left striatum (tumor area) were collected by an Olympus FV1000 microscope on a ×40 objective.

**4T1-luc breast cancer mouse model and oncolytic treatment**. All mice were purchased from the Experimental Animal Center of Huazhong Agricultural University. In order to construct a TNBC, 4T1-luc cells ($1 \times 10^6$) in 10 µl of RIPM1640 medium were implanted into the fat pad of the fourth pair of the left breast of Balb/c female mice or Nu/Nu nude female mice (6–8 weeks old).

To verify the oncolytic effect of rAAV-P2 on the TNBC mouse model, the animals were randomly divided into two groups intratumorally injected with

$1 \times 10^{10}$ vg of rAAV-P2 and rAAV-DIO-GSDMD$^{NT}$ in 10 µl of PBS 6 days post implantation, respectively (Fig. 4a). The mice were then kept in standard laboratory conditions and the size of long ($L$) and short ($W$) diameters of tumor from each mouse was recorded with a vernier caliper every 3–4 days. The total tumor volume of each animal was calculated according to the ellipsoid volume calculation formula (i.e., $1/2 \times L \times W^2$). The mice were sacrificed 24 days post-rAAV treatment. The tumors were photographed and weighed.

To verify the oncolytic effect of rAAV-P2 on the TNBC Nu/Nu nude mouse model, the animals were randomly divided into two groups intratumorally injected with $1 \times 10^{10}$ vg of rAAV-P2 and rAAV-DIO-GSDMD$^{NT}$ in 10 µl of PBS 6 days post implantation, respectively (Supplementary Fig. 12). The mice were then kept in standard laboratory conditions and the size of long ($L$) and short ($W$) diameters of tumor from each mouse was recorded with a vernier caliper every 3–4 days. The total tumor volume of each animal was calculated according to the ellipsoid volume calculation formula (i.e., $1/2 \times L \times W^2$). One hundred and fifty mg/kg dose of D-luciferin (Promega) was intraperitoneally injected into five mice in each group on 19 days post implantation. The small animal in vivo imager (PerkinElmer) was used to image the luciferase 15 min after injection. The imaging software (PerkinElmer) was used to quantitatively analyze the expression of luciferase in the mouse model. The mice were sacrificed 19 days post-rAAV treatment. The tumors were photographed and weighed.

In order to analyze the effect of rAAV-P2 on tumor-infiltrating lymphocytes, the mice were anesthetized and sacrificed to separate tumor masses 14 days post-rAAV-P2 or -rAAV-DIO-GSDMD$^{NT}$ treatments. The tumors were cut into pieces and ground and then filtered through a 70-µm cell sieve to obtain a single-cell suspension. Lymphocytes were separated with The Mouse Tumor-Infiltrating Tissue Lymphocyte Separation Kit (Solarbio). After washing twice with PBS containing 1% bovine serum albumin (FACS solution) and blocking with anti-mouse CD16/CD32 (1:50) for 30 min. The tumor-infiltrating lymphocyte suspensions were incubated with antibodies CD3-PE (1:100), CD4-FITC (1:100), CD8a-PacBlue (1:100), and CD49b-APC (1:100). After incubating for 1 h, these samples were washed once with FACS solution and analyzed by a flow cytometer Cytoflex LX (Beckman Coulter). In addition, the untreated tumors were sectioned and stained with anti-mouse CD3-PE (1:200) antibody for 1 h and Hoechst for 1 min. Confocal images were collected by an Olympus FV1000 microscope on a 40-fold objective.

In order to verify the oncolytic effect of rAAV-P2 treatment combined anti-PD-L1 antagonist (BMS202) on TNBC mouse model, and then the animals were randomly divided and intratumorally injected with $1 \times 10^{10}$ vg rAAV-P2 and PBS 7 days post implantation (Fig. 5a). In the rAAV-P2 combined BMS202 treatment group, the mice were intraperitoneally injected with BMS202 at a dose of 20 mg/kg for 7 consecutive days 8 day post-rAAV-P2 injection. In the BMS202 treatment group, the mice were intraperitoneally injected with BMS202 at a dose of 20 mg/kg for 7 consecutive days 8 days post-PBS injection. In the rAAV-P2 treatment group, the mice were intraperitoneally injected with PBS for 7 consecutive days 8 days post-rAAV-P2 injection. In the rAAV-P2 treatment group, the mice were intraperitoneally injected with PBS for 7 consecutive days 8 days post-rAAV-P2 injection. In the rAAV-P2 treatment group, the mice were intraperitoneally injected with PBS for 7 consecutive days 8 days post-rAAV-P2 injection. In the control group, the mice were intraperitoneally injected with PBS for 7 consecutive days 8 days post-PBS injection. The vernier caliper was used to measure the long diameter ($L$) and short diameter ($W$) of the tumor mass every 3–4 days. The tumor volume was calculated according to the ellipsoid volume calculation formula (i.e., $1/2 \times L \times W^2$). One hundred and fifty mg/kg dose of D-luciferin (Promega) was injected intraperitoneally into four mice in each group 21 days post implantation. The small animal in vivo imager (PerkinElmer) was used to image the luciferase 15 min post injection. The imaging software (PerkinElmer) was used to quantitatively analyze the expression of luciferase in the mouse model. At 30 days post-tumor implantation, the mice were sacrificed by anesthesia, and the tumors were photographed and weighed. In addition, propidium iodide was used to detect pyroptosis in the tumors according to the previous study[6].

To verify the oncolytic effect of Ad5-P2 on the TNBC mouse model, the animals were randomly divided into three groups intratumorally injected with $1 \times 10^8$ PFU of Ad5-P2, $1 \times 10^{10}$ vg of rAAV-P2 and PBS 6 and 14 days post implantation, respectively (Supplementary Fig. 18a). The mice were kept in standard laboratory conditions and the size of long ($L$) and short ($W$) diameters of tumor from each mouse was recorded with a vernier caliper every 3–4 days. The total tumor volume of each animal was calculated according to the ellipsoid volume calculation formula (i.e., $1/2 \times L \times W^2$). One hundred and fifty mg/kg dose of D-luciferin (Promega) was injected intraperitoneally into four mice in each group 28 days post implantation. The small animal in vivo imager (PerkinElmer) was used to image the luciferase 15 min post injection. The imaging software (PerkinElmer) was used to quantitatively analyze the expression of luciferase in the mouse model. The mice were sacrificed 28 days post implantation. The tumors were photographed and weighed.

**Evaluation of tumor targeting**. All mice were purchased from the Experimental Animal Center of Huazhong Agricultural University. In order to construct a TNBC, 4T1 cells ($1 \times 10^6$) in 10 µl of RIPM1640 medium were implanted into the fat pad of the fourth pair of the left breast of Balb/c female mice (6–8 weeks old). To examine whether the intratumoral injection of rAAV is restricted to

transducing genes in the tumor, the animals were randomly divided into two groups injected intratumorally with $1 \times 10^{10}$ vg of rAAV-CMV-luc and PBS (Supplementary Fig. 7), respectively. One hundred and fifty mg/kg dose of D-luciferin (Promega) was intraperitoneally injected into each mouse 15 days post implantation. The small animal in vivo imager (PerkinElmer) was used to image the luciferase 15 min after injection.

**scRNA-sequencing**. The tumor-infiltrating lymphocytes isolated as described in "FACS analyses of tumor-infiltrating lymphocytes." The CD45+ immune cells were enriched using a Beckman Coulter FACS MofloXDP flow cytometer. Cell viability was monitored in real-time during the preparation of the single CD45+ immune-cell suspension. Around 5000 cells from each experimental group were barcoded and pooled using the BD Rhapsody™ device. Samples were prepared following the manufacturer's protocol and sequenced by Illumina NovaSeq 6000 sequencer. The BD Rhapsody WTA Local bioinformatics pipeline was used for sample demultiplexing, barcode processing, alignment, filtering, UMI counting, and aggregation of the sequencing runs. For quality control of the scRNA-sequencing procedure, the cells with <300 genes and the cells with transcript counts for mitochondria-encoded genes more than 10% of the total transcript counts were removed from subsequent analyses. Genes detected in fewer than three cells across the dataset were also excluded, yielding a preliminary expression matrix of 5159 cells. After obtaining the digital gene expression data matrix, Seurat (v.3.2.3, Resolution = 1) was used for dimension reduction, clustering, and differential gene expression analyses. The clusters were annotated via SingleR and manually checked by canonical markers. The aggregated cell–cell communication network was visualized by CellChat (1.1.0). The number of interactions or the total interaction strength between any two cell groups was shown by a circle plot. The differential number of interactions or interaction strength in greater detail was shown by heatmap. All the significant interactions (L–R pairs) from MDSC, macrophage, and CD8+ T cell group to other cell groups were analyzed using netVisual_chord_gene.

**Transcriptome analysis of 4T1-luc breast cancer mouse model**. In order to analyze the effect of rAAV-P2 treatment on the tumors of the TNBC mouse model, the mice were anesthetized and sacrificed to separate the tumors 14 days post-rAAV-P2 or -rAAV-DIO-GSDMD$^{NT}$ treatment. After extracting the total RNA from the tumor tissue, the bulk RNA-seq libraries were constructed by using RNA-seq Lib Prep Kit for Illumina (ABclonal) and subjected to high-throughput sequencing (Genewiz). The upstream data were analyzed using the nf-core rnaseq pipeline[36]. The parameters "--aligner hisat2--skipBiotypeQC--genome mm10" were used. DEseq2[37] was used for differential gene expression analysis and exported the normalized count matrix. The DEGs were enriched with GO by the GO-BiologicalProcess-EBI-UniProt-GOA-ACAP-ARAP-08.08.2020 library by Cytoscape/ClueGo software. The filtering conditions were as follows, $p \leq 0.00001$, Correction Method Used = Bonferroni step down, Min GO Level = 7, Max GO Level = 15, and Kappa Score Threshold = 0.5.

The DEGs in the positively regulated T cell proliferation (GO:0042102) signaling pathway were uploaded to the STRING website (https://string-db.org/) to download a tab-separated value (.TSV) file. The log$_2$ (fold change) values of these DEGs were then imported into the Cytoscape software to generate interactive network diagrams.

**Reporting summary**. Further information on research design is available in the Nature Research Reporting Summary linked to this article.

## Data availability

scRNA-seq and bulk RNA-seq data that support the findings of this study have been deposited at the National Genomics Data Center (NGDC) under accession numbers PRJCA005365 and PRJCA003564. Statistical analysis of article-related data has been performed using the GraphPad Prism (version 8) software. Source data are provided with this paper. The remaining data are available within the Article, Supplementary information, or Source data file. Source data are provided with this paper.

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

## Acknowledgements

We thank Professor Jiahuai Han, Professor Jingping Ge, Professor Mingqian Feng, Professor Anding Zhang, Professor Guiqing Peng, and Professor Fuqiang Xu for reagents; Professor Feng Shao (National Institute of Biological Sciences, Beijing) for helpful comments on the manuscript; Professor Zhenfang Fu (University of Georgia) for helpful polishing of the manuscript; Dagang Tao, Xi Zhang, Xueli Jiang, Nan Zhang, Weijia Zhang, Yanyan Zou and Xiaojian Cao for technical assistance. This research was supported by the Fundamental Research Funds for the Central Universities (Grant No. 2662018PY025, 2662019YJ004, 2662018PY028 to G.C., J.L., and S.L.), the National Natural Science Foundation of China (Grant No. 819723082 to G.C.) and Key Frontier Projects of Applied Fundamentals of Wuhan Science and Technology Bureau (Grant No. 2019020701011438 to G.C.).

## Author contributions

G.C. conceived the project, designed the experiments, and wrote the manuscript. S.H. and J.L. conceived the project. S.L. discussed the results and commented on the manuscript. Y.L. and W.H. performed the majority of the experiments and collected the data and contributed to manuscript preparation. X.H., X.L., and Y.H. assisted with the animal studies. Z.H. provided technical assistance. Y.L., X.G., and W.X. analyzed the data. K.K. assisted with manuscript preparation.

## Competing interests

The authors declare no competing interests.
