## [Peer Review File · Nature Communications]

Reviewers' Comments:

Reviewer #2:

Remarks to the Author:

In this study, Lu Y, et. al, developed two novel approaches to package N-terminal gasdermin domain (GSDMNT) which is extreme cytotoxic into AAV vectors. They demonstrated the AAV-GSDMNT is capable to induce pyroptosis, temporarily open blood-brain barrier and recruit the tumor infiltrating lymphocytes into brain. The combined AAV-GSDMNT and anti-PD-L1 treatment further improved the oncolytic effect. Overall, I found this work interesting, however some points that should be addressed before publication.

In general, the two-vector platform, AAV-Cre and AAV- DIO- GSDMNT, is not ideal for translational applications. More efforts should be spent on optimizing the sf9-AAV-GSDMNT.

Comments:

Figure 1. The efficacy of mCBA promoter in mammalian cells should be tested. In my knowledge, CAG promoter and CBA promoter is similar. The difference between promoters should be illustrated.

Figure 2. AAV produced from sf9 cells and mammalian cells may vary in the transduction efficacy. Plasmid transfection may be able to figure out if the vector genome or capsid led the differences on pyroptosis.

Figure 3. The intratumor injection of AAV-GSDMNT showed therapeutic benefits. If the tumor cells were bilateral planted and the AAV vectors were intratumorally injected into one side, will the treatment still beneficial?

Figure 4. what is the vector genome distribution in the tumor tissues? Since GSDMNT is toxic, are the adjacent tissue and other tissues transduced by AAV?

Figure 5. Tumor infiltrating lymphocytes should be examined.

Minor:

The rationale of selecting AAV2-DJ in this study, not other AAV serotypes, should be explained.

Reviewer #3:

Remarks to the Author:

The manuscript by Lu et al provides datasets to assert oncolytic adeno-associated virus (AAV) vectors can be used for tumor therapy by targeted expression of N-terminal gasdermin domain. While the manuscript provides a cohesive description of the work, there are important questions in the rationale, interpretation of data, and limitations of the study.

The very term "oncolytic viruses" implicates the ability of virus/vector to replicate within the tumor cells so upon causing cell death other cells will be infected by the agent for serial lysis in a bystander effect. Recombinant AAV lacks this important ability and even wild-type AAV cannot replicate in a target cell unless the cell is co-infected by a helper virus.

There is not comparison between oncolytic vectors like adenovirus to show this approach is superior to clinically tested oncolytic approach for tumors.

Use of cre-system is technically exciting, but why not use an inducible promoter that can be used both for vector packaging and for in vivo application?

Anti-tumor immune effects as presented are very limited to support the interpretation. Performing RNA-Seq from whole tumor cells does not identify what cell(s) contributed to the variation in immune signatures. Single cell sequencing and follow-up studies by adoptive transfer of identified immune effector(s) to pre-existing tumors would have answered some of the claims.

The manuscript can be improved by a careful reading and editing for grammar errors.

Reviewer #4:

Remarks to the Author:

In this MS by Yuan lu et al., the authors have studied the role and translational applicability of the gasdermin D mediated pyroptosis in cancer immunotherapy. The authors developed an adenoviral system to express and regulate the biologically active form of N-terminal fragment of gasdermin D protein, which induces robust pyroptotic cell death in target cells, by membrane damage. They demonstrated that this system is not leaky, less toxic when not induced, and induces robust cell death in both in vitro and in vivo cancer model systems. The authors also provide convincing data showing that their oncolytic adeno-associated virus (oAAV) expressing GSDMNT promotes tumor regression, associated with T cell infiltration, and further completed by the anti-PD-L1 treatment-mediated checkpoint therapy. Overall, the study is well-designed and executed. The data supports the central findings conclusively.

Specific comments

Although, it is interesting to see that GSDMNT changes the TME and promotes recruitment of T cells, it is not clear if the tumor regressionism is substantially complemented by the T cells? It would be needed to further validate these assumptions using T cell deficient mouse models.

While the findings reported for the GSDMNT are very convincing, it is not clear if the cancer cell death mediated by the GSDMNT fragment is uniquely capable of inducing an immunogenic cell death and tumor regression or not? In other words, it is important to test another gasdermin (Eg: GSDMA or GSDMB) to validate the concept of pyroptosis as the key mechanism driving the immunogenic lytic cell death and tumor regression.

Reviewer #2 (Remarks to the Author): with expertise in AAV

In this study, Lu Y, et. al, developed two novel approaches to package N-terminal gasdermin domain (GSDMNT) which is extreme cytotoxic into AAV vectors. They demonstrated the AAV-GSDMNT is capable to induce pyroptosis, temporarily open blood-brain barrier and recruit the tumor infiltrating lymphocytes into brain. The combined AAV-GSDMNT and anti-PD-L1 treatment further improved the oncolytic effect. Overall, I found this work interesting, however some points that should be addressed before publication. In general, the two-vector platform, AAV-Cre and AAV- DIO- GSDMNT, is not ideal for translational applications. More efforts should be spent on optimizing the sf9-AAV-GSDMNT.

Response: Thank you very much for your valuable comments.

Comments:

Figure 1. The efficacy of mCBA promoter in mammalian cells should be tested. In my knowledge, CAG promoter and CBA promoter is similar. The difference between promoters should be illustrated.

Response: Thanks for this comment. As suggested, we performed a sequence alignment of mCBA promoter, CBA promoter and CAG promoter. As shown in Fig.1, the mCBA promoter contains an extra sequence in the 5-terminal region and a couple of SNPs in the “CMV enhancer”, “chicken beta actin promoter” regions, and a truncation “chimeric intron” region. These differences in the mCBA promoter might be one of the reasons for its inactivity in insect Sf9 cells. At the same time, we constructed plasmids containing CAG, CAG or mCBA promoter to drive eGFP expression, and then analyzed the averaged fluorescence intensity. As shown in Fig.2, the average fluorescence intensities of the cells transfected with mCBA promoter were stronger than that with CBA promoter, but weaker than that with CAG promoter. This result indicates the mCBA promoter has moderate activity in mammalian cells and is suitable for the expression of GSDM^{NT} in mammalian cells.

Fig. 1: The sequence alignment of CAG, CBA and mCBA promoters.

Fig.2: Transcription activity assay of CAG, CBA and mCBA promoter activity in HEK 293T, HeLa and 4T1-luc cells. a, Representative fluorescence microscopy of HEK 293T and HeLa cells showing the expression of eGFP driven by different promoters at 48 h post-transfection. Scale bars, 20 μ m. **b,** Quantitative results of the average fluorescence intensity of eGFP expression driven by different promoters in HEK 293T, HeLa and 4T1-luc cells. Mean \pm s.d., two-tailed unpaired Student's t-test. All data are representative of three independent experiments.

Figure 2. AAV produced from sf9 cells and mammalian cells may vary in the transduction efficacy. Plasmid transfection may be able to figure out if the vector genome or capsid led the differences on pyroptosis.

Response: Thanks for this comment. As suggested, we constructed pAAV-mCBA-DIO-GSDMD^{NT} and pAAV-ef1 α -DIO-GSDMD^{NT} plasmids, and co-transfected them into HEK 293T, HeLa and 4T1-luc cells together

with pAAV-Cre plasmids respectively and then measured pyroptosis ratio by the LDH-Glo Cytotoxicity Assay. As shown in Fig.3c-d, compared with the pAAV-ef1 α -DIO-GSDMD^{NT} and pAAV-Cre co-transfection group, the pAAV-mCBA-DIO-GSDMD^{NT} and pAAV-Cre co-transfection group had fewer pyroptosis cells, indicating that the mCBA promoter activity is weaker than ef1 α Promoter. This data suggests that the lower pyroptosis efficiency of rAAV-P1 than that of rAAV-P2 is related to the weaker activity of the mCBA promoter.

To further explore the underlying mechanism of the different induced pyroptosis efficiency between these two systems, we used HEK 293T cells and Sf9 cells to package rAAV/293-CMV-eGFP and rAAV/sf9-CMV-eGFP respectively, and then analyzed the average fluorescence intensities of these two viruses infected cells. As shown in Fig.3a-b, the average fluorescence intensities of the cells infected by rAAV/293-CMV-eGFP with same titer were stronger than that of the rAAV/sf9-CMV-eGFP. These results indicate that the rAAV packaged by insect sf9 cells has lower transduction efficiency, which might be related to the lower affinity of insect-derived capsid proteins for mammalian cells. Thus the relatively lower efficiency of rAAV-P1 mediated pyroptosis could be also due to the lower transduction efficiency of rAAV packaged by sf9 cells.

Fig. 3: The comparison of induced pyroptosis efficiency between different rAAVs and the versatility of rAAV-P2 packaging strategy. **a**, Representative fluorescence microscopy of HEK 293T, HeLa, and 4T1-luc cells showing the expression of eGFP transduced by rAAV produced in insect Sf9 cells (rAAV/sf9-CMV-eGFP) and mammalian 293 cells (rAAV/293-CMV-eGFP) 48 hours post infection, respectively. Scale bars, 20 μ m. **b**, Quantitative results of the average fluorescence intensity of the eGFP expressed by rAAV/sf9-CMV-eGFP and rAAV/293-CMV-eGFP in HEK 293T, HeLa and 4T1-luc cells, respectively. Mean \pm s.d., two-tailed unpaired Student's t-test. **c**, Image of the cells transfected with different plasmids as indicated. Arrows indicate pyroptotic cells. Scale bars, 20 μ m. **d**, Comparison of LDH release-based cell death assay in the cells transfected with different plasmids as indicated. Mean \pm s.d., two-tailed unpaired Student's t-test.

Figure 3. The intratumor injection of AAV-GSDMNT showed therapeutic benefits. If the tumor cells were bilateral planted and the AAV vectors were intratumorally injected into one side, will the treatment still beneficial?

Response: Thanks for this inspiring comment. As suggested, we constructed a Wistar rat glioma model with bilateral tumor inoculation. 7 days post tumor implantation, the rats were intratumorally injected with rAAV-P2 and rAAV-DIO-GSDMD^{NT} in left side of the brain, respectively. 14 days post the injection of rAAV-P2, the rats were anesthetized and the brains were taken for paraffin sections and H&E staining (Fig.4a). As shown in Fig.4b-c, compared with the rAAV-DIO-GSDMD^{NT} control rats, the size of the bilateral tumors were both significantly reduced. Moreover, obvious necrosis were observed in bilateral brains tumor tissues with the rAAV-P2 treatment. These data showed that the rAAV-P2 induced pyroptosis can systematically enhance the anti-tumor immunity in the whole brain, which has also been reported by other oncolytic viruses¹.

Fig. 4: rAAV-P2 treatment of GBM rat model with bilateral tumor inoculation. **a**, Experimental timeline for rAAV-P2 treatment of GBM rat model with bilateral tumor inoculation. **b**, H&E staining of brain slices of two groups of rats 21 days post tumor implantation. Scale bar, 2,000 μ m. **c**, The left and right brain tumor areas of the two groups of rats on the 21st day post tumor implantation. Mean \pm s.d., one-tailed unpaired Student's t-test.

Figure 4. what is the vector genome distribution in the tumor tissues? Since GSDMNT is toxic, are the adjacent tissue and other tissues transduced by AAV?

Response: Thanks for this important comment. To verify the distribution of rAAV, we constructed an orthotopic TNBC mouse model using 4T1 cells and intratumorally injected the tumor with rAAV-CMV-luc expressing luciferase. As shown in Fig.5, the vast majority of the luciferin signal was detected in the tumor mass but not other organs by both in vivo and postmortem imaging. This data indicated that rAAV injected intratumorally is mostly limited in the solid tumor. Therefore, intratumoral injection of rAAV-P1 or rAAV-P2 in the solid tumor will very unlikely lead to the risk of pyroptosis in other tissues.

Fig.5: Evaluation of tumor targeting of AAV intratumor injection. a, Schematic of AAV-CMV-luc treatment on TNBC mouse model. **b,** Luciferase imaging of mouse in each group 9 days post AAV-CMV-luc or PBS intratumor injection. **c,** Luciferase imaging of tumors and main organs in each group 9 days post rAAV-CMV-luc or PBS intratumor injection. All data are representative of two independent experiments.

Figure 5. Tumor infiltrating lymphocytes should be examined.

Response: Thanks for this comment. As suggested, we analyzed the tumor-infiltrating lymphocytes of an orthotopic breast cancer mouse model treated with rAAV-P2 and PD-L1 inhibitor by flow cytometry. As shown in Fig.6, compared with the PBS and PD-L1 control group, the tumors of the mice in the rAAV-P2 treatment group and the rAAV-P2 combined with PD-L1 inhibitor treatment group contained significantly more CD4⁺, CD8⁺ T cells and NK cells. This data indicates that the rAAV-P2 treatment and rAAV-P2 combined with PD-L1 inhibitor treatment can induce the recruitment of tumor-infiltrating lymphocytes.

Fig.6: Anti-PD-L1 therapy boosts the effect of rAAV-P2 treatment on TNBC in mouse model and lymphocytes infiltrating analysis. **a**, Schematic of rAAV-P2 combined with anti-PD-L1 treatment on TNBC mouse model. **b**, Representative flow cytometry plots for 4T1-luc tumor infiltrating CD3⁺ cells analysis. **c-f**, Quantification of the tumor infiltrating lymphocytes from TNBC mouse model. $n = 5$ mice for PBS and rAAV-P2, 4 mice for anti-PD-L1 and rAAV-P2 + anti-PD-L1. Mean \pm s.e.m., one-tailed unpaired Welch's t-test.

Minor:

The rationale of selecting AAV2-DJ in this study, not other AAV serotypes, should be explained.

Response: Thanks for this comment. We selected AAV2-DJ in this study because of its broad-spectrum characteristics. It is convenient to study the treatment of tumor derived from different tissues. In clinical tumor treatment, we can select more specific AAV serotypes for different tumors from different tissues to achieve higher tumor targeting efficiency.

Reviewer #3 (Remarks to the Author): with expertise in AAV and cancer immunotherapy

The manuscript by Lu et al provides datasets to assert oncolytic adeno-associated virus (AAV) vectors can be used for tumor therapy by targeted expression of N-terminal gasdermin domain. While the manuscript provides a cohesive description of the work, there are important questions in the rationale, interpretation of data, and limitations of the study.

The very term “oncolytic viruses” implicates the ability of virus/vector to replicate within the tumor cells so upon causing cell death other cells will be infected by the agent for serial lysis in a bystander effect. Recombinant AAV lacks this important ability and even wild-type AAV cannot replicate in a target cell unless the cell is co-infected by a helper virus.

Response: Thanks for your valuable comments. To be more precise, we have rephrased the “oncolytic viruses” to “recombinant AAV” in the manuscript.

There is not comparison between oncolytic vectors like adenovirus to show this approach is superior to clinically tested oncolytic approach for tumors.

Response: Thanks for this comment. As suggested, we also developed a packaging strategy for the expression of GSDMD^{NT} in oncolytic adenovirus vector. Since the AdMax packaging system of adenoviral vectors contains Cre recombinase already, we then selected Flp recombinase for this packaging strategy to avoid the expression of GSDMD^{NT} (Fig.8a). Ad5-Flp and Ad5-fDIO-GSDMD^{NT} were packaged (Fig.8b) and mixed (the mixture called Ad5-P2), and then tested on HEK293, Hela and 4T1-luc cells. As shown in Fig7c-d, Ad5-P2 can induce the pyroptosis in HEK293, Hela and 4T1-luc. This result indicates that this packaging strategy can be successfully applied for the expression of GSDMD^{NT} in adenovirus vector (Fig.7). The therapeutic effect of Ad5-P2 was then evaluated in an orthotopic triple-negative breast cancer mouse model. 6 days and 14 days post tumor implantation, PBS, Ad5-P2 and rAAV-P2 were injected into the tumor of the mouse model, respectively (Fig.9a). As shown in Fig.9, Ad5-P2 treatment inhibited tumor growth in the TNBC mice model, suggesting that Ad5-P2-mediated pyroptosis can be also applied for tumor treatment. In theory, Ad5-P2 should have a better therapeutic effect on tumors than rAAV-P2. However, our data shows that there is no significant difference in the therapeutic effects between these two systems (Fig.9b-g). This might be because the pyroptosis treatment effect is stronger and thus overwhelm the oncolytic effect mediated by Ad5 itself or the AAV system has better expression efficiency.

Fig.7: The flow diagram of Ad5-P2 packaging. 1) Construction of plasmid pHBAd-fDIO-GSDM^{NT}. 2) Co-transfect with “pBHGloxΔE1, 3Cre” into HEK 293 cells to package Ad5-fDIO-GSDM^{NT}. FRT-STOP-FRT can avoid the expression of GSDM^{NT}. 3) Monoclonal by agarose plaque selection, amplified by serial passage on HEK 293 cells. 4) After CsCl density gradient centrifugation and 5) Concentration, Ad5-fDIO-GSDM^{NT} was obtained. 6) Ad5-Flp is also packaged to delete STOP for GSDM^{NT} expression. In this way, it can avoid the expression of GSDM^{NT} during virus packaging.

Fig.8: The Strategies for packaging Type 5 adenovirus vector (Ad5) expressing GSDMD^{NT} and the oncolytic effect analysis. **a**, Schematic of the strategy using Flp/FRT system to package Ad5-fDIO-GSDMD^{NT}. Co-infection of AAV-Cre can delete STOP (Ad5-P2) to induce pyroptosis. **b**, Representative images of fluorescent plaques formed by Ad5-Flp or Ad5-fDIO-GSDMD^{NT} infected HEK 293 cells. Scale bars, 20 μm . **c**, Images of HEK 293, HeLa and 4T1-luc cells infected with Ad5-P2 (arrows indicate pyroptotic cells). Scale bars, 20 μm .

Fig. 9: Ad5-P2 treatment on TNBC in mouse model. **a**, Schematic of Ad5-P2 or rAAV-P2 treatment on TNBC mouse model. **b**, Luciferase imaging of 4T1-luc breast tumors 21 days post tumor implantation. **c**, Corresponding quantification of luciferase expression in **b**. Mean \pm s.e.m., from four independent replicates, one-tailed unpaired Student's t-test. **d**, Tumor volume of an individual mouse. n = 6 mice per group. **e, g**, Average tumor volume (**e**) and weight (**g**) of mice as indicated. Mean \pm s.e.m., two-tailed unpaired Student's t-test. **f**, Photographs of representative tumors 28 d post tumor implantation.

Use of cre-system is technically exciting, but why not use an inducible promoter that can be used both for vector packaging and for in vivo application?

Response: Thanks for this inspiring comment. As suggested, we used the classical Tetracycline-inducible promoter to avoid gene expression of GSDM^{NT} during packaging (Fig.10a). However, it was very difficult to obtain high-yield pAAV-rtTA-CMV-TRE-GSDM^{NT} plasmid, even by increasing the amount of bacteria (Fig.10b). This may be due to the poor state of *E. coli* caused by the leaky expression of the tetracycline inducible promoter, suggesting that the plasmid may also have leaking expression in HEK 293T cells (Fig.10c). To verify the above hypothesis, we transfected pAAV-rtTA-CMV-TRE-GSDM^{NT} in HEK 293T cells. As shown in Fig.10d, in the pAAV-rtTA-CMV-TRE-GSDM^{NT} transfection group, most of the cells showed pyroptosis. This data shows that the strategy using a tetracycline inducible promoter is very hard to package large amount of rAAV-GSDM^{NT}, as GSDM^{NT} is an extremely strong pyroptosis inducing protein and a very low-level leaky expression is enough to kill the cells. Nevertheless, this great idea could be still feasible by extensively testing various inducible promoters.

Fig.10: Packaging rAAV-GSDM^{NT} with a tetracycline inducible promoter. **a**, The strategy of Tetracycline-inducible promoter packaging rAAV-GSDM^{NT}. **b**, The yield of two different plasmids, Mean ± sem, two-tailed unpaired Student's t-test. **c**, The missed expression of plasmid pAAV-rTA-CMV-TRE-GSDMDNT leads to HEK 293T cell pyroptosis. **d**, The images of HEK 293T cells transfected with different plasmids (arrows indicate pyroptotic cells).

Anti-tumor immune effects as presented are very limited to support the interpretation. Performing RNA-Seq from whole tumor cells does not identify what cell(s) contributed to the variation in immune signatures. Single cell

sequencing and follow-up studies by adoptive transfer of identified immune effector(s) to pre-existing tumors would have answered some of the claims.

Response: Thanks for this comment. To further study the effect of rAAV-P2 treatment on the tumor microenvironment (TME) at single cell level, we performed scRNA-Seq analysis on 4T1 tumor infiltrating lymphocytes treated with rAAV-P2 (Fig.11a-d). The results of scRNA-Seq analysis showed that rAAV-P2 treatment can increase the proportion of CD4⁺, CD8⁺ T cells and NK cells in the tumor (Fig.11e, f), indicating that rAAV-P2 treatment can recruit more lymphocytes cells. By performing ligand-receptor analysis on single cell data, we found that, By performing ligand-receptor analysis on single cell data, we found that the number of ligand-receptor interactions between the immunosuppressive cell population (MDSC) and other cell populations was decreased in the rAAV-P2 treatment group (Fig.11g and Fig.12a). Moreover, the interaction strength was weaker after treatment (Fig.12a). Whereas, the cell-to-cell communication interaction strength and number of the plasmacytoid Dendritic Cells (pDC) and CD8⁺ T cells to other cells all increased (Fig.12). The results of the differential gene analysis of each cell cluster in the scRNA-seq showed that rAAV-P2 therapy can down-regulate the expression levels of the protumoural and immunosuppressive genes (*S100a9*, *S100a8*, *Cxcl2* and *Csf1*), and up-regulate gene expression level of the proinflammatory chemokine genes (*Ccl5*, *Ccl8*, *Ccl9* and *Cxcl10*), T and/or natural killer cell activation genes (*Klrl1*, *Havcr2* and *Ld2*) or effector genes (*Gzma*, *Gzmb* and *Gzmk*) in tumor infiltrating lymphocytes (Fig.13). These data indicate that rAAV-P2 treatment can stimulate the anti-tumor immune response in the tumor, thereby facilitate killing the tumor. Although we have identified some immune effectors, it is still technically challenging at this moment to harvest and transfer these effector cells with proper expression of specific pro- and anti-immune genes and block the MDSC simultaneously in the tumor. Nevertheless, this could be a potential highly safe approach for tumor therapy in the near future.

Fig.11: The single-cell transcriptome analysis of tumor infiltrating lymphocytes of TNBC mouse model treated with AAV-P2. **a**, Schematic of rAAV-P2 treatment on TNBC mouse model. **b**, Gating strategy and representative flowcytometry plots for the enrichment of 4T1 tumor-infiltrating single CD45⁺ immune cells. **c**, Heat map of nine immune-cell clusters with unique signature genes. Colours on top of the map indicate the immune-cell clusters. The four or five marker genes used for each cluster are listed alongside the cluster. **d**, Signature gene-expression patterns for the corresponding cell clusters on the t-SNE plot (n = 4 mice, 5,159 cells). **e**, t-SNE plots of tumor infiltrating single CD45⁺ immune cells of 4T1 tumors from mice treated with rAAV-control (rAAV-ef1 α -DIO-GSDMD^{NT}) and rAAV-P2, respectively. **f**, The relative frequencies of different clusters. **g**, Condition-specific linkages between MDSC ligands and other lymphocyte cluster receptors in rAAV-DIO-GSDMD^{NT} treatment group and rAAV-P2 treatment group.

a**b****c**
Fig.12: Illustrations of the integrated analysis of inter- and intracellular signaling. **a,** The heatmap of differential number of interactions or interaction strength among different cell populations. The top-colored bar plot represents the sum of column of values displayed in the heatmap (incoming signaling). The right-colored bar plot represents the sum of row of values (outgoing signaling). **b,** Condition-specific linkages between CD8⁺ T cell ligands and other lymphocyte cluster receptors in rAAV-DIO-GSDMD^{NT} treatment group and rAAV-P2 treatment group. **c,** Condition-specific linkages between pDC ligands and other lymphocyte cluster receptors in rAAV-DIO-GSDMD^{NT} treatment group and rAAV-P2 treatment group.

Fig.13: The single-cell transcriptome differential genes analysis of tumor infiltrating lymphocytes of TNBC mouse model treated with rAAV-P2. a-d, The expression levels of protumoural and immunosuppressive genes (a), proinflammatory chemokine (b) and T and/or natural killer cell activation (c) or effector (d) genes in each immune cell cluster.

The manuscript can be improved by a careful reading and editing for grammar errors.

Response: Thanks for this comment. We have carefully checked through the manuscript and also asked an English native speaker to polish the manuscript.

Reviewer #4 (Remarks to the Author): with expertise in gasdermins and pyroptosis - and cancer

In this MS by Yuan lu et al., the authors have studied the role and translational applicability of the gasdermin D mediated pyroptosis in cancer immunotherapy. The authors developed an adenoviral system to express and regulate the biologically active form of N-terminal fragment of gasdermin D protein, which induces robust pyroptotic cell death in target cells, by membrane damage. They demonstrated that this system is not leaky, less toxic when not induced, and induces robust cell death in both in vitro and in vivo cancer model systems. The authors also provide convincing data showing that their oncolytic adeno-associated virus (oAAV) expressing GSDMNT promotes tumor regression, associated with T cell infiltration, and further completed by the anti-PD-L1 treatment-mediated checkpoint therapy. Overall, the study is well-designed and executed. The data supports the central findings conclusively.

Thanks for your valuable comments.

Specific comments

Although, it is interesting to see that GSDMNT changes the TME and promotes recruitment of T cells, it is not clear if the tumor regressionism is substantially complemented by the T cells? It would be needed to further validate these assumptions using T cell deficient mouse models.

Response: Thanks for this comment. As suggested, we investigated the tumor regression effect on an orthotopic triple-negative breast cancer in athymic Nu/Nu mouse model (Fig.14a). As shown in Fig.14b-f, no therapeutic effect of rAAV-P2 treatment were observed in the athymic Nu/Nu mouse model. This data suggests that the recruitment of T cells is crucial for the therapeutic effect of rAAV-P2.

Fig.14: The rAAV-P2 treatment on TNBC Nude mouse model. **a**, Schematic of rAAV-P2 treatment in TNBC nude mouse model. **b**, Luciferase imaging of 4T1-luc breast tumors 19 d post tumor implantation. **c**, Corresponding quantification of luciferase expression in **b**. Mean \pm s.e.m., from five independent replicates, two-tailed unpaired Student's t-test. **d**, Average tumor volume of mice as indicated. $n = 6$ mice for PBS, AAV-DIO-GSDMD^{NT} (rAAV-ef1 α -DIO-GSDMD^{NT}) and AAV-P2. Mean \pm s.e.m., two-tailed unpaired Student's t-test. **e**, Photographs of representative tumors 19 d post treatment. **f**, Average tumor weight of mice as indicated. Mean \pm s.e.m., two-tailed unpaired Student's t-test.

While the findings reported for the GSDMNT are very convincing, it is not clear if the cancer cell death mediated by the GSDMNT fragment is uniquely capable of inducing an immunogenic cell death and tumor regression or not? In other words, it is important to test another gasdermin (Eg: GSDMA or GSDMB) to validate the concept of pyroptosis as the key mechanism driving the immunogenic lytic cell death and tumor regression.

Response: Thanks for this comment. To verify the versatility of our packaging rAAV-GSDM^{NT} strategy and the pyroptosis effect of other GSDM genes, we chose GSDMB^{NT} and GSDME^{NT} for rAAV-GSDM^{NT} packaging, and obtained rAAV-P2B and rAAV-P2E. As shown in Fig.15a, rAAV-P2B and rAAV-P2E were packaged successfully, and induced obvious pyroptosis in HEK 293T, Hela, and 4T1-luc cells. Meanwhile, we also found that compared with rAAV-P2, rAAV-P2B and rAAV-P2E have relatively lower efficiency in mediating pyroptosis (Fig.15b), which might be related to the low efficiency of GSDMB^{NT} and GSDME^{NT} in cell membrane pore formation. These data demonstrate that our strategy can be also be used to package rAAV expressing all GSDM^{NT} or other cytotoxic genes.

Fig. 15: Packaging and pyroptosis analysis of rAAV-DIO-GSDMB^{NT} and GSDME^{NT}. **a**, Image of the cells infected with rAAV-DIO-GSDMB^{NT}, rAAV-DIO-GSDME^{NT}, rAAV-P2B (the mixture of rAAV-DIO-GSDMB^{NT} and rAAV-Cre) and rAAV-P2E (the mixture of rAAV-DIO-GSDME^{NT} and rAAV-Cre), respectively. Arrows indicate pyroptotic cells. Scale bars, 20 μm. **b**, Comparison of LDH release-based cell death assay in the cells infected

with different rAAVs. Mean \pm s.d., two-tailed unpaired Student's t-test. All data are representative of three independent experiments.

- 1 Russell, L. *et al.* PTEN expression by an oncolytic herpesvirus directs T-cell mediated tumor clearance. *Nat Commun* **9**, 5006, doi:10.1038/s41467-018-07344-1 (2018).

Reviewers' Comments:

Reviewer #2:

None

Reviewer #3:

Remarks to the Author:

The revised manuscript by Lu et al has incorporated several new figures and changes to the text. However, the outcome of new studies, with oncolytic adenovirus vector in direct comparison to the recombinant AAV vector does not yield any superiority of the latter. This brings an important question as to the applicability of this system in cancer therapy. Oncolytic adenoviruses have failed in the clinic and how, apart from commendable technical incorporations and experimental variables, this work is significant is questionable.

Using PD-L1 inhibitor appears more like a timely popularity rather than high scientific rationale.

There are several places where the experiments were done in a cursory manner than paying attention to understanding and presenting tumor immune characteristics in the mouse model used.

Reviewer #4:

Remarks to the Author:

The authors responded well and have provided substantial data to support their claims.

Reviewer #5:

Remarks to the Author:

In the manuscript from Yuan Lu, Wenbo He and co-authors, entitled "Novel strategies to package recombinant Adeno-Associated Virus expressing GSDMNT for tumor treatment", among other experiments, the authors performed single cell RNA sequencing of tumor infiltrating lymphocytes. I have a few questions related to the single cell RNA data analysis/.

1) There is a possible disparity between Figure 4f and and supplementary 9. The expression of some genes, such as Cd19 are not where it would be expected based on the cluster assignment, as in the supplementary it is located at the same position of cluster 6. While in figure 4f B cells are cluster 9. Could you please check this?

2) I suggest inclusion of additional plots (heatmaps, scatter plots or violin plots), of canonical markers with a more detailed explanation which markers they identify. I believe Neutrophils are miss-assigned (they might be pro-inflammatory macrophages), pDCs are most likely mDCs. CD4 T cells are likely pDCs. MDSC need to be better evaluated as well. One suggestion for the authors would be to use automated notation (such as SingleR), with manual double checks using canonical markers.

3) Supplementary 10 has a few interactions with capital gene names. Are these human genes? Can you explain if these interactions used only mouse, or a mouse-human hybrid source of reference?

Reviewer #3 (Remarks to the Author):

The revised manuscript by Lu et al has incorporated several new figures and changes to the text. However, the outcome of new studies, with oncolytic adenovirus vector in direct comparison to the recombinant AAV vector does not yield any superiority of the latter. This brings an important question as to the applicability of this system in cancer therapy. Oncolytic adenoviruses have failed in the clinic and how, apart from commendable technical incorporations and experimental variables, this work is significant is questionable. Using PD-L1 inhibitor appears more like a timely popularity rather than high scientific rationale. There are several places where the experiments were done in a cursory manner than paying attention to understanding and presenting tumor immune characteristics in the mouse model used.

Response: Thanks for the comments. In theory, Ad5-P2 should have a better therapeutic effect on tumors than rAAV-P2. However, our data showed that there was no significant difference between these two systems. This might be because the pyroptosis has extremely strong cell death effect and thus overwhelm the oncolytic effect mediated by Ad5 itself. Other than this, the AAV system might have better expression and infection efficiency and could possibly induce stronger pyroptosis effect.

One advantage of AAV system is that specific rAAV targeting can be achieved by incorporating high affinity ligands into the viral vector particles, such as Her2-rAAV, to realize specific gene transference into Her2/neu⁺ positive tumor. Furthermore, the progress in tumor-specific promoters could also provide alternative solutions for the GSDM^{NT} tumor targeting. With the progress of in these fields, we believe these GSDM^{NT} system may have great potential in cancer therapy.

Immune checkpoint blockade therapy such as anti-PD1/PD-L1 has been successfully applied in clinical treatments. However, this approach is only effective in certain type of cancers, probably due to the distinct immunosuppressive TME. Our RNA-seq and tumor-infiltrating lymphocyte analysis data showed that rAAV-P2 treatment can up-regulate the expression of chemokine and cytokine-related genes and increase lymphocyte infiltration in the tumors. Consistent with previous studies, our data revealed that the pyroptosis induced by GSDM^{NT} can alter the immunosuppressive TME. Additionally, the up-regulation of the immune checkpoint (PD1/PD-L1) genes implies that anti-PD-L1 therapy can enhance oncolytic effect of rAAV-GSDMD^{NT}. Accordingly, our data demonstrated that the combination of anti-PD-L1 therapy and rAAV-P2 can indeed significantly

improve the oncolytic effect of rAAV-P2 and may have a promising prospect in anti-tumor immunotherapy.

As the main goal of our study is to develop novel strategies to circumvent the cytotoxicity of GSDMD^{NT}, efficiently produce and deliver GSDM^{NT} into tumor cells and pave the way for the oncolytic therapy of cancer via pyroptosis, we didn't focus on the delineation of the detailed tumor immune characteristics. Nevertheless, we agree with the reviewer that it would be important to further explore the molecular mechanisms of this PD-L1 and pyroptosis mediated cancer therapy in the future study.

Reviewer #5 (Remarks to the Author): with expertise in scRNAseq, immunology

In the manuscript from Yuan Lu, Wenbo He and co-authors, entitled “Novel strategies to package recombinant Adeno-Associated Virus expressing GSDMNT for tumor treatment”, among other experiments, the authors performed single cell RNA sequencing of tumor infiltrating lymphocytes. I have a few questions related to the single cell RNA data analysis.

1) There is a possible disparity between Figure 4f and and supplementary 9. The expression of some genes, such as Cd19 are not where it would be expected based on the cluster assignment, as in the supplementary it is located at the same position of cluster 6. While in figure 4f B cells are cluster 9. Could you please check this?

Response: As suggested by the reviewer, we carefully checked the figures and found that the gene expression patterns on the t-SNE plot of Cd19 in Supplementary Figure 9C was indeed accidentally mislabeled. This mistake has been corrected in the revised figures (Fig.1c). We apologize for this mistake and really appreciate the reviewer's careful checking.

2) I suggest inclusion of additional plots (heatmaps, scatter plots or violin plots), of canonical markers with a more detailed explanation which markers they identify. I believe Neutrophils are miss-assigned (they might be pro-inflammatory macrophages), pDCs are most likely mDCs. CD4 T cells are likely pDCs. MDSC need to be better evaluated as well. One suggestion for the authors would be to use automated notation (such as SingleR), with manual double checks using canonical markers.

Response: Thanks for this comment. As suggested, we re-analyzed the single cell RNA data

with higher resolution for clustering and annotated the cells by both SingleR and canonical cell markers. After obtaining the digital gene-expression data matrix, Seurat (v.3.2.3, Resolution = 1) was used for dimension reduction and clustering. The clusters were firstly annotated by SingleR, and then manually double checked with canonical markers. At the same time, we also added more violin plots to show the detailed description of the marker genes of each cell cluster (Fig.1d). The major conclusion regarding the alteration of the TME is basically the same. The annotation of the re-classified cell clusters was updated in the revised manuscript.

Fig. 1: The single-cell transcriptome analysis of tumor infiltrating lymphocytes of TNBC mouse model treated with rAAV-P2. The single-cell transcriptome analysis of tumor infiltrating lymphocytes of TNBC mouse model treated with rAAV-P2. a, Gating strategy and representative flowcytometry plots for the enrichment of 4T1 tumor-infiltrating single CD45⁺

immune cells. **b**, Heatmap of ten immune-cell clusters with unique signature genes. Colours on top of the map indicate the immune-cell clusters. The four or five marker genes used for each cluster are listed alongside the cluster. **c**, Signature gene-expression patterns for the corresponding cell clusters on the t-SNE plot (n = 4 mice, 5,159 cells). **d**, Violin plots show the expression of gene markers from the corresponding cell clusters. The cell cluster corresponding to each marker gene is marked in red.

3) Supplementary 10 has a few interactions with capital gene names. Are these human genes? Can you explain if these interactions used only mouse, or a mouse-human hybrid source of reference?

Response: Thanks for this comment. We used mouse source of reference only for the interaction analysis. As the capital gene name usually refers to human gene in cell interaction analysis, we have changed all the capital gene name in the revised manuscript (Fig.2).

Fig. 2: Illustrations of the integrated analysis of inter- and intracellular signaling. a-c, Condition-specific linkages between MDSC (a), CD8⁺ T cell (b) and Macrophage (c) ligands and other lymphocyte cluster receptors in rAAV-DIO-GSDMD^{NT} treatment group and rAAV-P2 treatment group.

Reviewers' Comments:

Reviewer #5:

Remarks to the Author:

The authors have adequately addressed the comments I raised and I am happy with the increased robustness and reproducibility achieved by those changes.